# Hybrid Federated Learning for Feature & Sample Heterogeneity: Algorithms and Implementation

**Xinwei Zhang**\*  
*Department of Electrical and Computer Engineering*  
*University of Minnesota*  
*zhan6234@umn.edu*

**Wotao Yin**  
*Department of Mathematics*  
*University of California, Los Angeles*  
*wotaoyin@math.ucla.edu*

**Tianyi Chen**†  
*Department of Electrical and Computer Engineering*  
*Rensselaer Polytechnic Institute*  
*chent18@rpi.edu*

**Mingyi Hong**\*  
*Department of Electrical and Computer Engineering*  
*University of Minnesota*  
*mhong@umn.edu*

**Reviewed on OpenReview:** *https://openreview.net/forum?id=qc2lmWkvk4*

## Abstract

Federated learning (FL) is a popular distributed machine learning paradigm dealing with distributed and private data sets. Based on the data partition pattern, FL is often categorized into horizontal, vertical, and hybrid settings. All three settings have many applications, but hybrid FL remains relatively less explored because it deals with the challenging situation where *both* the feature space and the data samples are *heterogeneous*. Hybrid FL combines the advantages of both horizontal and vertical FL, addressing some of their individual limitations, such as the same-features requirement of the former and the same-entities requirement of the latter.

This work designs a novel mathematical model that allows clients to aggregate distributed data with heterogeneous and possibly overlapping features and samples. Our main idea is to partition each client's model into a feature extractor part and a classifier part, where the former can be used to process the input data, while the latter is used to perform the learning from the extracted features. The heterogeneous feature aggregation is done by building a server model that assimilates local classifiers and feature extractors through a carefully designed matching mechanism. A communication-efficient algorithm is then designed to train both the client and server models. Finally, we conducted numerical experiments on multiple image classification data sets to validate the performance of the proposed algorithm. To our knowledge, this is the first formulation and algorithm developed for hybrid FL.

## 1 Introduction

Federated Learning (FL) is an emerging distributed machine learning (ML) framework that enables heterogeneous clients – such as organizations or mobile devices – to collaboratively train ML models (Konečný

---

\*The work of M. Hong and X. Zhang was partially supported by NSF grant EPCN-2311007 and CNS-2003033. This work is also part of AI-CLIMATE: "AI Institute for Climate-Land Interactions, Mitigation, Adaptation, Tradeoffs and Economy," and is supported by USDA National Institute of Food and Agriculture (NIFA) and the National Science Foundation (NSF) National AI Research Institutes Competitive Award no. 2023-67021-39829.

†The work of T. Chen was partially supported by National Science Foundation Grant 2047177.

Table 1: Examples of applications that generate heterogeneous data

| Application | Client | Feature Blocks | Sample |
|---|---|---|---|
| Medical Diagnosis | clinic | output of different diagnostic devices | patient |
| Recommendation System | retailer | record of different product categories | customer |
| Social Network | SNS provider | user activity & relationship | SNS user |

et al., 2016; Yang et al., 2019). The development of FL aims to address practical challenges in distributed learning, such as feature and data heterogeneity, high communication cost, and data privacy requirements.

The challenge due to heterogeneous data is particularly evident in FL. The most well-known form of heterogeneous data is *sample heterogeneity* (SH), where the distributions of training samples are different across the clients (Kairouz et al., 2021; Bonawitz et al., 2019). Severe SH can cause common FL algorithms such as FedAvg to diverge (Khaled et al., 2019; Karimireddy et al., 2020b). Recently, better-performing algorithms and system architectures for distributed ML (including FL) under SH include Karimireddy et al. (2020b); Li et al. (2018); Wang et al. (2020); Fallah et al. (2020); Vahidian et al. (2021).

Besides SH, another form of heterogeneity is *feature heterogeneity* (FH). Traditionally, we say the samples are FH if we can partition them into subsets that bear distinct features. In the FL setting, when the sample subsets of different clients have different, but not necessarily distinct, features, we call it FH. That is, under FH, different clients have unique and possibly also common features. FH and SH arise in ML tasks such as collaborative medical diagnosis (Ng et al., 2021), recommendation systems (Yang et al., 2020), and graph learning (Zhang et al., 2021), where the data collected by different clients have different and possibly overlapping features and sample IDs. Next, we provide a few examples.

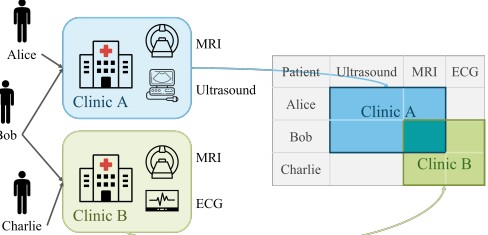

Figure 1: The heterogeneous data distribution in a medical diagnosis example.

*Medical diagnosis application* (see Figure 1). The clients are clinics, and they collect data samples from patients. Each clinic may have a *different set* of diagnostic devices, e.g., clinic A has MRI and ultrasound, while clinic B has MRI and electrocardiographs (ECG). FH arises as the feature set of each sample collected by clinic A may *partially overlap* with that done by clinic B. Besides FH, SH also arises as multiple clinics may not have the chance of treating the same patient and each patient usually visits only a subset of clinics.

*Recommendation systems (Yang et al., 2020; Zhan et al., 2010).* In this case, the clients are large retailers, and they collect samples (such as shopping records) from their customers. The retailers share a subset of common products and a subset of common customers.

A third example pertains to an application of learning over *multiple social networks* (Zhang et al., 2021; Guo & Wang, 2020). Here the clients are social network providers (e.g., Twitter, Facebook), and the samples are the set of participating users, their activities and relations. We summarize these three examples in Table. 1.

In the previous three applications, client data can be heterogeneous in *both* feature and sample. Surprisingly, none of the existing FL algorithms can fully handle such data. Rather, *Horizontal* FL (HFL) and *Vertical* FL (VFL) methods can handle data with only one heterogeneity, the former with SH and the latter with FH. By keeping only the common features (and ignoring the other features), we can avoid FH and apply an HFL method. By keeping only the common samples (and discarding the remaining samples), we can avoid SH and apply a VFL method. Clearly, they both waste data.

Consider the HFL algorithms (Konečný et al., 2016; Karimireddy et al., 2020b;a; Dinh et al., 2021). The clients perform multiple local model updates, and the server averages those updates and broadcasts the new model to the clients. This scheme works when the clients share the same model and their data share an identical set of features (see Figure 2b for an illustration); otherwise, the server cannot average their models.

Consider the *Vertical* FL (VFL) algorithms (Liu et al., 2019; Chen et al., 2020). They split the model into blocks. Each client processes a subset of the blocks while the server aggregates the processed features

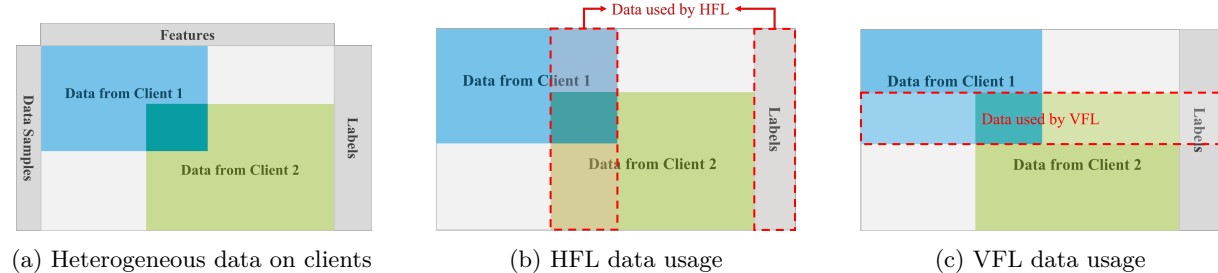

(a) Heterogeneous data on clients        (b) HFL data usage        (c) VFL data usage

Figure 2: The data distribution patterns of a) heterogeneous client data; b) HFL and c) VFL.

to compute training losses and gradients. They require all the clients to have the same set of samples (see Figure 2c); otherwise, they cannot compute the loss and its gradient.

According to Yang et al. (2019); Rahman et al. (2021), the FL setting with heterogeneous features and samples is referred to as *hybrid FL*. To develop a hybrid FL method, we must address the following challenges:

1. **Global and local inference requires global and local models.** Hybrid FL makes it possible for a client to make its local inference and also for all the clients (or the server) to make a global inference. The former requires only the features local to a client; the latter requires all the features and training a global model at the server.

2. **Limited data sharing.** In typical HFL, the clients do not share their local data or labels during training. In VFL, the labels are either made available by the clients to the server (Chen et al., 2020) or stored in a designated client (Liu et al., 2019). A hybrid FL system may be subject to a "no sharing" requirement, so it is desirable to develop a method in which the server has no access to any data, including the labels.

3. **Sample synchronization.** A technical challenge with VFL is that the server wants the clients to draw the same mini-batch of samples at each iteration. This challenge is exacerbated in hybrid FL since not all the clients will have the same samples. Therefore, to avoid idling clients, a hybrid FL method should allow uncoordinated sample draws.

**Our contributions:** Towards addressing the previous challenges, this work proposes a novel model and its training method. We summarize our contribution as follows.

1. We propose a new hybrid FL approach. For each client, the model consists of a feature extractor and a subsequent classifier. The clients collaborate and share their knowledge through building a model at the server that assimilates local classifiers and features. The assimilation is achieved by a matching mechanism inspired by the non-parametric modeling idea in Yurochkin et al. (2019). This approach enables both global and local inferences and can handle data with both SH and FH. To our knowledge, this is the first concrete hybrid FL model in the literature.

2. We develop a hybrid FL algorithm that enables knowledge transfer among the clients. The algorithm maintains data locality, so the server does not access clients' data, and it allows uncoordinated sample draws by the clients.

3. We evaluated the performance of the hybrid FL algorithm on a number of real datasets. The learned model achieved an accuracy that was comparable to that of a centrally trained model.

## 1.1 Related work

**Federated graph learning** (FGL) is applied to molecular classification (He et al., 2021), relation or node classification for social networks (Zhang et al., 2021; Ng et al., 2021) and financial network (Suzumura et al., 2019). In the first application, the graphs are relatively small and the clients have largely many graphs (Zhang et al., 2021; He et al., 2021). In the last two application scenarios, the clients possess partial yet overlapping data of a single large graph, including partial node and edge information (Zhang et al., 2021). However,

existing FGL algorithms mainly focus on the first application scenario (He et al., 2021) and fail to deal with the latter two scenarios. So we cannot apply them to our hybrid FL setting.

**HFL** has a popular algorithm FedAvg (Konečnỳ et al., 2016), which adopts the computation-then-aggregation strategy. The clients locally perform a few steps of model updates, and then the server aggregates the updated local models and averages them before sending the updated global model back to the clients. Beyond model averaging, PFNM (Yurochkin et al., 2019) and FedMA (Wang et al., 2020) use a parameter-matching-based strategy and FedGKT (He et al., 2020) uses a knowledge distillation strategy to get better global model performance, and they do not require the global model to have the same size as the local models. All HFL algorithms assume their data have the same set of features.

**Personalized FL (PFL)** has been studied as a potential way to tackle different levels of task heterogeneity. MAML (Jiang et al., 2019; Fallah et al., 2020) uses meta-learning to build a global model that can fast adapt to heterogeneous data distribution; FedProx (Li et al., 2018) and LG-FedAvg (Hanzely & Richtárik, 2020) regularize the distance between the local models and the global model. MOCHA (Smith et al., 2017) and FedU (Dinh et al., 2021) combine multi-task learning with FL to train models for personalized tasks. FedPer (Arivazhagan et al., 2019) separates the model into base+personalized layers to decouple the common and personal knowledge. However, most of the algorithms assume that all local models take the same input size and format.

**Vertical Federated Learning.** In VFL, the features and thus the models are divided over different clients (Hardy et al., 2017; Ma et al., 2019; Liu et al., 2019; Chen et al., 2020). VFL has fewer results than HFL. Federated Block Coordinate Descent (FedBCD) (Liu et al., 2019) uses a parallel BCD-like algorithm to optimize the local blocks and transmits essential information for the other clients to compute their local gradients. Vertical Asynchronous Federated Learning (VAFL) (Chen et al., 2020) assumes that the server holds the global inference model while local clients train the feature extractors that deal with the local features. VFL algorithms ignore the overlapping features and can only make joint inferences, which requires full client participation.

**Federated Contrastive Learning (FedCL).** FedCL is another set of algorithms related to Hybrid FL. In this setting, the clients hold non-overlapping features and partially overlapping samples and aim to learn separate models for local inference (Kang et al., 2022; He et al., 2022). The algorithms perform vertical FL on the overlapping samples to train a global guidance model and perform local contrastive learning (self-supervised learning) with the non-overlapped local data to train local models. Compared with our setting, the FedCL algorithm requires overlapping samples and transmitting intermediate features as VFL and fails to make use of overlapping feature spaces and non-overlapping samples during VFL training.

## 2 Problem Formulation

In this section, we first provide a mathematical characterization of the heterogeneous data distributions of interest to this work. We then propose a unified hybrid FL model.

**Notation:** Due to the nature of hybrid FL, we must carefully set up its notation. We denote the all one (column) vector of length $d$ as $\mathbb{1}_d$; the identity matrix of size $d$ as $I_d$; the positive integer set $\{1, 2, \ldots, N\}$ as $[N]$. Feature selection below uses a selector matrix of dimension $d_1 \times d_2$, which belongs to the following set: **Data description:** See Figure 3a for an illustration of a dataset with three clients, where the client datasets has no *fully* overlapped sample or feature, so neither HFL nor VFL can be used. We consider a hybrid FL system with $M$ clients indexed by $m \in [M]$, and they collaborate to accomplish the same task. For convenience, we index the server as $m = 0$.

First, assume that each sample can have at most $d_0$ feature blocks, and the $i$th block has the set $\mathcal{D}_i$ of features, $i \in [d_0]$; client $m$ has a set of $d_m$ feature blocks indexed by $\mathcal{I}_m$, that is, we write $\langle \mathcal{D}_{i_m} \rangle_{i_m \in \mathcal{I}_m}$ and write the feature space of client $m$ as $\mathcal{X}_m = \prod_{i_m \in \mathcal{I}_m} \mathcal{D}_{i_m}$, which is a Cartesian product of the subset of the feature blocks possessed by client $m$. Similarly, we denote the "full feature" space as $\mathcal{X}_0 = \prod_{i=1}^{d_0} \mathcal{D}_i$, which is the Cartesian product of all feature blocks.

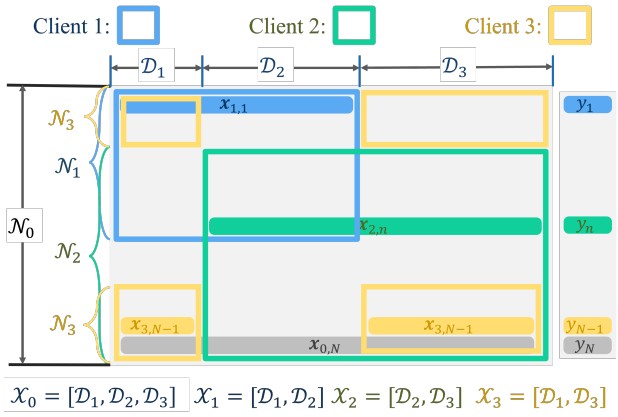

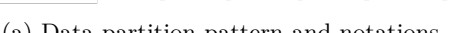

$\mathcal{X}_0 = [\mathcal{D}_1, \mathcal{D}_2, \mathcal{D}_3]$   $\mathcal{X}_1 = [\mathcal{D}_1, \mathcal{D}_2]$   $\mathcal{X}_2 = [\mathcal{D}_2, \mathcal{D}_3]$   $\mathcal{X}_3 = [\mathcal{D}_1, \mathcal{D}_3]$

(a) Data partition pattern and notations.

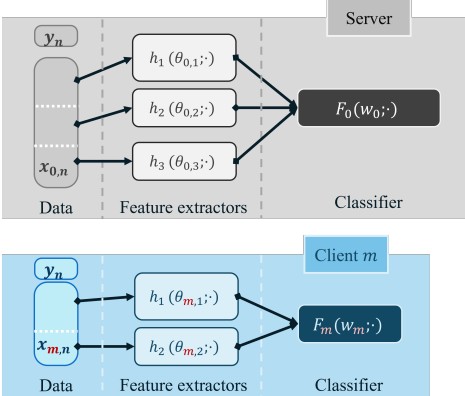

(b) Block structure of client and server models

Figure 3: The partitioned data and notations, and the structure of the client and server models with heterogeneous feature extractors and classifiers.

Second, client $m$ holds a private dataset with index set $\mathcal{N}_m$ and the samples $(x_{m,n}, y_n)$ for $n \in \mathcal{N}_m$, where $x_{m,n} \in \mathcal{X}_m$ denotes the features of the $n^{\text{th}}$ sample on client $m$, and $y_n$ denotes the label of the $n^{\text{th}}$ sample. Collecting all the clients' data together, we can define the (virtual) global dataset to have sample index set $\mathcal{N}_0 = [N]$, with samples $(x_{0,n}, y_n)$ where $x_{0,n} \in \mathcal{X}_0$ denotes the "full feature" of the $n^{\text{th}}$ sample (for the precise relation between the full-featured $x_{0,n}$ and the local sample $x_{m,n}$, please see the property P2 below).

The dataset defined above satisfies the following properties.

**P1)** The global index set is the union of the clients' index sets:

$$\mathcal{N}_0 = \bigcup_{m=1}^{M} \mathcal{N}_m, \quad \text{which implies} \quad \mathcal{N}_m \subseteq \mathcal{N}_0.$$

**P2)** For a given client $m$, the features of the $n^{\text{th}}$ sample is a sub-vector of the "full features". That is, there exists a selector matrix $P_m$ such that it can map the global feature $x_{0,n}$ to $x_{m,n}$:

$$x_{m,n} = P_m x_{0,n}, \quad \text{for some} \quad P_m \in \mathcal{S}(d_m, d_0), \tag{1}$$

where $P_m$ is a selector matrix that selects the feature blocks on client $m$ from the full feature.

*Remark 1.* (Data structures of HFL and VFL) The data structure that the HFL deals with can be viewed as a special case of what has been described above, where the clients have fully overlapping features, i.e., $d_0 = 1$, $P_m = 1$, $\forall m \in [M]$; Similarly, the data structure for VFL can be viewed as a special case that the clients have fully overlapping sample indices, i.e., $\mathcal{N}_m = \mathcal{N}_0$, $\forall m \in [M]$. □

**Model design:** With the above description of data, we are ready to present the proposed hybrid FL model and the corresponding optimization problem.

**Client and server model design:** Similar to VFL, we split the ML model into *feature extractors* and *classifiers*. Each feature extractor takes a feature block as input and extracts an intermediate feature as output; the classifier takes the concatenated intermediate features of multiple feature extractors as input and outputs the prediction.

As illustrated by Figure 3b, on client $m$, the feature extractor $h_{i_m}(\theta_{m,i_m}; \cdot)$ for input feature block $\mathcal{D}_{i_m}$ is parameterized by $\theta_{m,i_m}$ for all $i_m \in \mathcal{I}_m$. The feature extractors can have different neural network architectures (e.g., CNN for CT/MRI images, LSTM/Transformer for medical records, and 1-D CNN for ECG data). We denote the concatenated feature extractors and their parameters as:

$$H_m(\Theta_m; \cdot) := [h_{i_m}(\theta_{m,i_m}; \cdot)]_{i_m \in \mathcal{I}_m}, \quad \text{and} \quad \Theta_m := [\theta_{m,i_m}]_{i_m \in \mathcal{I}_m}. \tag{2}$$

The classifier $F_m(w_m; \cdot)$ is parameterized by $w_m$, and we denote the prediction loss function as $\ell(\cdot, \cdot)$. The data processing procedure on client $m$ is described as follows:

**(1)** The features $x_{m,n}$ of the $n^{\text{th}}$ sample are passed to the feature extractors $\{h_{i_m}(\theta_{m,i_m}; \cdot)\}_{i_m \in \mathcal{I}_m}$;

**(2)** The classifier $F_m(w_m; \cdot)$ makes the prediction based on the concatenated output of the feature extractors $H_m(\Theta_m; x_{m,n})$;

**(3)** The prediction $F_m(w_m; H_m(\Theta_m; x_{m,n}))$ and the true label $y_n$ together evaluates the loss $\ell(\cdot, \cdot)$.

With the specified data processing procedure, the *prediction loss* on client $m$ is defined as:

$$f_m(\Theta_m, w_m) := \frac{1}{|\mathcal{N}_m|} \sum_{n \in \mathcal{N}_m} \ell(F_m(w_m; H_m(\Theta_m; x_{m,n})), y_n). \tag{3}$$

Additionally, the server will have a model with *full* feature extractors, concatenated with a classifier; see the top figure in Fig. 3b. This model structure covers a wide range of ML models for classification and regression problems, e.g., image classification, language processing, and recommendation systems.

*Remark 2.* (Local and server models). A few remarks are ready. First, although the construction of the client model has been partly motivated by the model splitting idea from VFL, one key difference with VFL is that each client holds a *complete* model, capable of performing local inference *without* communication to the server. Second, it is important to have a separate server model, because: **1)** in case a test data with "full feature" comes in, the server can deal with it; **2)** in case a new client comes who needs to process a new subset of features, it can directly download the corresponding feature extractors from the server, which significantly reduces the complexity of building the local model; and most importantly **3)** the server's model is instrumental in helping the clients to learn from each other's data (as we will see shortly). □

At this point, we have defined the models and the prediction loss for each individual client. A key question is: how the client can effectively collaborate and leverage each other's data to train high-quality server/client models? Unlike HFL, where all clients share the same model, the clients in this problem have local models (i.e., feature extractors and classifiers) of different sizes to deal with feature heterogeneity. Therefore, one cannot directly perform the conventional model averaging.

**Model matching:** To enable effective collaboration among the clients, our idea is to properly *match* different parts of the model, by imposing a number of carefully designed regularizers.

First, it is natural to assume that when client $m$ and $m'$ share the same feature block $\mathcal{D}_i$, the corresponding feature extractors $h_i(\theta_{m,i}; \cdot)$ should produce the same output, that is $\theta_{m,i} \approx \theta_{0,i} \approx \theta_{m',i}$. Therefore, we impose the following regularizer for the feature extractors, which matches the $i^{\text{th}}$ feature extractor at user $m$ with the corresponding extractor at the server:

$$r_{m,1}(\Theta_m, \Theta_0) := \sum_{i \in \mathcal{I}_m} \frac{1}{2} \|\theta_{m,i} - \theta_{0,i}\|^2 = \frac{1}{2} \|\Theta_m - P_m \Theta_0\|^2, \tag{4}$$

where $P_m$ is the data selection matrix defined in (1) and $\Theta_m$ concatenates parameters defined in (2).

We then design the regularizer for the classifiers. As the classifiers on different clients share partially overlapping input and identical output space, we model the client's classifiers $w_m$ as some "pruned" versions of the server-side classifier $w_0$, but with *unknown* pruning pattern. More specifically, assume that $w_m \in \mathbb{R}^{d_{m,w}}, w_0 \in \mathbb{R}^{d_{0,w}}$, we impose the following regularizer for the classifier:

$$r_{m,2}(w_m, \Pi_m, w_0) = \frac{1}{2} \|w_m - \Pi_m w_0\|^2, \quad \text{s.t.} \quad \Pi_m \in \mathcal{S}(d_{m,w}, d_{0,w}), \tag{5}$$

where $\Pi_m$ is a selection matrix defining the unknown pruning pattern. It is important to note that, the pruning pattern matrices $\Pi_m$'s are unknown and need to be optimized. On the contrary, when in the definition of the feature extractor regularizer (4), the matrix data selection matrices $P_m$'s are fixed, and they are defined by the data partitioning pattern. Detailed discussion about the structure of the constraints on the pruning matrix $\Pi_m$'s and the regularizer $r_{m,2}$ are given in Appendix A.1

**Overall problem formulation:** By combining the models discussed in the previous two subsections, we arrive at the following training problem:

$$\min_{\{\Theta_m, w_m\}_{m=0}^M, \{\Pi_m\}_{m=1}^M} \sum_{m=1}^M p_m \left( f_m(\Theta_m, w_m) + \mu_1 \cdot r_{m,1}(\Theta_m, \Theta_0) + \mu_2 \cdot r_{m,2}(w_m, \Pi_m, w_0) \right), \tag{6}$$
$$\text{s.t.} \quad \Pi_m \in \mathcal{S}(d_{m,w}, d_{0,w}), \ \forall m \in [M],$$

where $\mu_1, \mu_2$ are hyper-parameters for the regularizers; $p_m$'s are the weights for each local problem satisfying $\sum_{m=1}^M p_m = 1$, with common choices $p_m = \frac{1}{M}$ or $p_m = \frac{|\mathcal{N}_m|}{|\mathcal{N}|}$.

*Remark 3.* (Relation with HFL). When $d_0 = 1$, that is, there is only a single feature block across all the clients, then the data structure can be handled by the conventional HFL. Below let us discuss the relations between our model (6) and some popular HFL models. First note that when $d_0 = 1$, the feature extractor regularizer (4) reduces to $r_{m,1}(\Theta_m, \Theta_0) = \frac{1}{2} \|\Theta_m - \Theta_0\|^2$.

1) **Reduction to FedMA (Wang et al., 2020) and Sub-FedAvg (Vahidian et al., 2021).** If we set $\Theta_m = I$, i.e., the features are directly processed by the $w_m$'s, then (6) is equivalent to the problem solved by FedMA and Sub-FedAvg.

2) **Reduction to FedProx (Li et al., 2018) and LG-FedAvg (Hanzely & Richtárik, 2020).** By setting $w_m = I.p_m = \frac{1}{M}$ and letting $\Theta_m$ directly predict the labels, the problem reduces to

$$\min_{\{\Theta_m\}_{m=0}^M} \frac{1}{M} \sum_{m=1}^M (f_m(\Theta_m) + \mu_1 \cdot r_{m,1}(\Theta_m, \Theta_0)), \tag{7}$$

which is equivalent to the formulation solved by FedProx and LG-FedAvg.

3) **Reduction to FedAvg.** Further by letting $\mu_1 \to \infty$ in (7), the regularizer enforces $\Theta_m$'s to achieve exact consensus, the problem reduces to the one solved by FedAvg.

4) **Reduction to FedPer (Arivazhagan et al., 2019).** By letting $\mu_2 = 0$ and $\mu_1 \to \infty$ in (6), the regularizer on $w_m$'s is removed and $\Theta_m$'s achieve exact consensus. In this case, $\Theta_m$ serves as the base layers while $w_m$'s are the personalized layers, equivalent to the model design of FedPer. □

*Remark 4.* (Relation with VFL). VFL assumes that the clients cannot perform prediction independently, so it directly trains a global model with the local data (Liu et al., 2019; Chen et al., 2020). In contrast, we assume that each client has sufficient features for independent training and construct a local model, which is further used to construct a global model. This way, we avoid data sharing and sample synchronization issues that often limit VFL use in practice. □

## 3 Algorithm Design

In this section, we propose a training algorithm for the proposed Hybrid FL formulation (6). This algorithm will alternate between the server-side updates and the client-side updates. To proceed, we will first split (6) into a server-side problem and a client-side problem, and then develop algorithms to optimize each part. One key consideration in our algorithm design is to ensure that the server-side model is optimized without directly accessing any clients' data.

**Problem splitting:** Notice that the problem contains parameter blocks $\{\Theta_m\}_{m=1}^M$, $\{w_m\}_{m=1}^M$, $\Theta_0$, $w_0$ and $\{\Pi_m\}_{m=1}^M$. First we divide the parameters into two groups: **1)** the server-side parameters $\Theta_0$, $w_0$, and $\{\Pi_m\}_{m=1}^M$ and **2)** the client-side parameters $\{\Theta_m\}_{m=1}^M$ and $\{w_m\}_{m=1}^M$.

By fixing the server-side parameters, (6) decomposes into $m$ independent problems, one for each client. The problem related to client $m$ is given by:

$$\min_{\Theta_m, w_m} f_m(\Theta_m, w_m) + \mu_1 \cdot r_{m,1}(\Theta_m, \Theta_0) + \mu_2 \cdot r_{m,2}(w_m, \Pi_m, w_0). \tag{8}$$

---

**Algorithm 1** Hybrid Federated Matching Algorithm (HyFEM)

---

1: **Input:** $w_0^0, \Theta_0^0, \{\Pi_m^0\}_{m=1}^M, \eta, T, Q, P$
2: **for** $t = 0, \ldots, T - 1$ **do**
3:    **for client** $m = 1, \ldots, M$ in parallel **do**
4:       $\Theta_m^{t,Q}, w_m^{t,Q} \leftarrow$ **ClientUpdate**$\left(\Theta_0^t, \Pi_m^t, w_0^t, Q, \eta\right)$       // *Local perturbed SGD solving (8)*
5:       Send client model $\Theta_m^{t,Q}, w_m^{t,Q}$ to server
6:    **for server do**
7:       $\Theta_0^{t+1} \leftarrow \left(\sum_{m=1}^M p_m P_m^T P_m\right)^{-1} \left(\sum_{m=1}^M p_m P_m^T \Theta_m^{t,Q}\right)$    //*E*xact minimization for (10)
8:       $w_0^{t+1}, \{\Pi_m^{t+1}\}_{m=1}^M \leftarrow$ **ModelMatching**$\left(\{w_m^{t,Q}, \Pi_m^t\}_{m=1}^M, P\right)$     // *S*olving (11)
9:       Distribute server model $w_0^{t+1}, \Theta_0^{t+1}, \{\Pi_m^{t+1}\}_{m=1}^M$ to clients
10: **Output:** $\{w_m^T, \Theta_m^T\}_{m=0}^M, \{\Pi_m^T\}_{m=1}^M$

---

Similarly, by fixing the client-side parameters, the $f_m$'s in (6) become constants, and the problem reduces to the following server-side problem:

$$\min_{\Theta_0, w_0, \{\Pi_m\}_{m=1}^M} \sum_{m=1}^M p_m \left(\mu_1 \cdot r_{m,1}(\Theta_m, \Theta_0) + \mu_2 \cdot r_{m,2}(w_m, \Pi_m, w_0)\right), \tag{9}$$
$$\text{s.t.} \qquad \Pi_m \in \mathcal{S}(d_{m,w}, d_{0,w}), \ \forall m \in [M].$$

The above problem can be naturally separated into two sub-problems. The first sub-problem is:

$$\min_{\Theta_0} \sum_{m=1}^M p_m \cdot r_{m,1}(\Theta_m, \Theta_0), \tag{10}$$

and the second one is:

$$\min_{w_0, \{\Pi_m\}_{m=1}^M} \sum_{m=1}^M p_m \cdot r_{m,2}(w_m, \Pi_m, w_0), \quad \text{s.t.} \quad \Pi_m \in \mathcal{S}(d_{m,w}, d_{0,w}), \ \forall m \in [M]. \tag{11}$$

**Algorithm design:** We propose a block coordinate descent type algorithm called Hybrid Federated Matched Averaging (HyFEM) in Algorithm 1 to solve (6) with the above problem splitting strategy and the sub-routines are given by Algorithm 2 in Appendix A.2. In global iteration $t$, the clients first perform $Q$ local perturbed SGD steps on problem (8) to optimize client models $w_m^t, \Theta_m^t$ (lines $1 - 7$ in Algorithm 2); then the server aggregates the updated client models, updates global feature extractors by optimizing (10) that has a closed-form solution as line 7 in Algorithm 1, and match the classifiers by optimizing (11); finally, the server distributes the models and the selection matrices to clients.

The major step in the algorithm is solving the sub-problem (11). We optimize it by the **ModelMatching** procedure described on lines $8 - 14$ of Algorithm 2 in Appendix A.2: **1)** for each client index $m'$, construct the server model $w_0^{t,p}$ without the impact of the selected client; 2) apply the Hungarian algorithm to solve a parameter assignment problem and obtain $\Pi_{m'}^{t,p+1}$ in at most $\mathcal{O}((d_{m,w})^3)$ run-time complexity Kuhn (1955). With a few rounds of updates, we obtain the server classifier and the selection matrices for each client. This procedure is inspired by the model matching algorithms Wang et al. (2020); Yurochkin et al. (2019) for matching parameters in deep neural networks of the same size. Our matching algorithm is a non-trivial extension to the existing model matching algorithm. Because the server-side and client-side models do not share the exact same functionality, we cannot replace the client-side models with the server-side model. Such a special property introduces some significant challenges for model matching. The detailed matching procedure is included in Appendix A.2.

*Remark 5.* Although Algorithm 1 seems to be complicated, it can be viewed as a problem with three parameter blocks $\mathcal{L}(\mathbf{x}, \mathbf{y}, \mathbf{z})$, where $\mathbf{x}$ is the collection of $\{w_m, \Theta_m\}_{m=1}^M$; $\mathbf{y}$ is the collection of $\{\Pi_m\}_{m=1}^M$ and $\mathbf{z}$ is $\{w_0, \Theta_0\}$. Then the update can be viewed as follows:

$$\underbrace{\mathbf{x}^+ \leftarrow \mathbf{x} - \eta \tilde{\nabla}_{\mathbf{x}} \mathcal{L}(\mathbf{x}, \mathbf{y}, \mathbf{z})}_{Q \text{ times}}, \ \mathbf{y}^+ \leftarrow \underset{\mathbf{y} \in \text{Range}(\mathbf{y})}{\arg\min} \mathcal{L}(\mathbf{x}^+, \mathbf{y}, \mathbf{z}), \ \mathbf{z}^+ \leftarrow \underset{\mathbf{z}}{\arg\min} \mathcal{L}(\mathbf{x}^+, \mathbf{y}^+, \mathbf{z}), \tag{12}$$

where $\tilde{\nabla}_{\mathbf{x}}\mathcal{L}(\cdot)$ denotes the stochastic partial gradient estimation w.r.t. $\mathbf{x}$. $\qquad\square$

**Theorem 1 (Informal)** *Suppose that for each $m \in [M]$, $f_m$ has Lipschitz continuous gradients w.r.t. $[\Theta_m, w_m]$ and that $w_0$ has a fixed dimension. Then with stepsize $\eta = \mathcal{O}(1/\sqrt{QT})$ and client update $Q = \mathcal{O}(T)$, by running Algorithm 1, the expected gradient norm square w.r.t. $\{\Theta_m, w_m\}_{m=1}^M$ converges with rate $\mathcal{O}(1/T)$ and the successive update difference $\left\|w_0^{t+1} - w_0^t\right\|^2 + \left\|\Theta_0^{t+1} - \Theta_0^t\right\|^2$ converges with rate $\mathcal{O}(1/T)$. Alternatively, if we assume the solution to $\{\Pi_m\}_{m=1}^M$ for sub-problem* (11) *is unique, and we update client models with one-step gradient descent, then Algorithm 1 asymptotically converges to the first-order stationary point of* (6)*.*

*Remark 6.* Theorem 1 is a non-trivial extension of the convergence results for traditional BCD-type algorithms. The major challenges in the analysis of HyFEM are: 1) it runs multiple, yet a fixed number of stochastic gradient updates on (potentially nonconvex) blocks $\{\Theta_m, w_m\}_{m=1}^M$, which results in non-strictly decrease; 2) the problem w.r.t. block $\{\Pi_m\}_{m=1}^M$ is nonconvex and non-smooth and does not have a unique global minimum. Such a setting is different from existing work on BCD-type algorithms. The detailed convergence statement and its proofs are given in Appendix B. $\qquad\square$

We highlight the merits of the proposed approach: **1)** Unlike the typical VFL formulations (Liu et al., 2019; Chen et al., 2020), our approach keeps the data at the clients. Hence, the local problems are fully separable. There is no sample-drawing synchronization needed during local updates; **2)** By utilizing the proposed model matching technique, we can generate a global model at the server, which makes use of full features. This makes the inference stage flexible: the clients can use either partial features (by using its local parameters $(\Theta_m, w_m)$) or the full features by requesting $(\Theta_0, w_0)$ from the server or letting the server do the inference.

Although we formulate the problem by adopting the idea of model splitting from VFL and model pruning/matching from HFL, optimizing (5) is still a non-trivial procedure. Specifically, we can only train clients' classifiers $w_m$'s of different sizes, and construct unknown server's classifier $w_0$ with $w_m$'s and find $\Pi_m$'s, while existing algorithms either require $w_0$ to be given (Vahidian et al., 2021), or $w_m$'s to have the same size (Wang et al., 2020; Yurochkin et al., 2019).

## 4 Numerical Experiments

To evaluate the proposed algorithms, we have conducted experiments on a number of standard datasets, and compared the results with several baselines including centralized training and stand-alone local training (without any client-server communication). Since existing FL algorithms cannot be applied to our setting where the client features are only partially overlapped, we do not compare HyFEM with other FL algorithms in this section. However, we include an additional set of experiments in Appendix C comparing HyFEM and FedProx with less heterogeneous features.

**Dataset & data splitting:** We consider the ModelNet40, Cifar-10, and EuroSAT datasets, the details of which are explained below. We also consider an additional multi-modal dataset, we refer the readers to Appendix C.3 for details.

**ModelNet40 (Wu et al., 2015):** ModelNet40 is a multiview object classification dataset that has 12 views from different angles as 12 feature blocks for each object. The dataset has $\mathcal{N}_0 = 40,000$ samples from 40 classes. **Cifar-10 (Krizhevsky, 2009):** Cifar-10 is an image classification dataset with $\mathcal{N} = 50,000$ samples from 10 classes. We manually split each image into (top left,top right,bottom left,bottom right)×(red,green,blue) blocks, resulting in total $d_0 = 12$ feature blocks. **EuroSAT (Helber et al., 2019):** EuroSAT is a land cover classification satellite image dataset with $\mathcal{N}_0 = 27,000$ samples from 10 classes, and the images are split into 12 feature blocks the same as Cifar-10.

In the training phase of each task, we manually assign a few feature blocks and classes to each client, so that the clients have partially overlapping features and samples and exhibit FH and SH. The settings are summarized in Table 2. It is worth pointing out that in setting ModelNet40:2, 12.08% of the data have *never* been used by any of the clients during training, and in all settings, there is no feature or sample that is shared by all clients, so VFL and HFL algorithms cannot be applied. We conduct two sets of experiments on ModelNet40 dataset where setting 1 uses $d_0 = 4$ views and setting 2 uses full $d_0 = 12$ views. The first setting has fewer features, so the classifiers are smaller and the matching procedure is easier and expected

Table 2: Experiment settings for each dataset. $d_0, d_m$ denote the # of feature blocks; $\mathcal{N}_0, \mathcal{N}_m$ denote the # of samples; $M$ denotes the number of clients.

| Dataset | $d_0$ | Classes | $\mathcal{N}_0$ | Client $M$ | $d_m$ | Classes/client | $\mathcal{N}_m$ |
|---|---|---|---|---|---|---|---|
| ModelNet40:1 | 4 | 40 | 40k | 4 | 3 | 20 | 20k |
| ModelNet40:2 | 12 | 40 | 40k | 8 | 6 | 15 | 15k |
| Cifar-10 | 12 | 10 | 50k | 9 | 6-8 | 5 | 25k |
| EuroSAT | 12 | 10 | 27k | 9 | 6-8 | 5 | 13.5k |

to be more accurate. Thus the performance of the server model should be closer to the model obtained with centralized training. In the second setting, the matching procedure is more complex than that of the first setting and should result in worse server model performance. The illustration of the data assignment pattern is given in Appendix C.2.

In the testing phase, the clients evaluate their model on all testing samples with corresponding feature blocks used in the training phase. We average over the accuracies obtained by the clients to obtain the averaged local accuracy. The global accuracy is evaluated using the matched server model on all testing samples with full features.

**Training settings:** In the experiments, we use the MLP model with one hidden layer as the classifier $f_m(w_m; \cdot)$. We use the CNN part of ResNet-18 followed by one pooling layer as the feature extractors for the Cifar-10 and EuroSAT datasets; we use the CNN part of ResNet-34 followed by one pooling layer as the feature extractors for ModelNet40 dataset. We use the following experiment settings as a comparison: **Centralized training:** we train a full-sized server model with all data. This setting serves as the performance upper bound among all trained models. **Stand-alone training:** each client trains a client model only with local data and without any communication. This setting serves as the baseline (and the performance lower bound) of HyFEM. In all settings, we fix the *total* number of updates (i.e., $T \cdot Q = 4096$, with $T = 128, Q = 32$) for fair comparison and tune the learning rate to achieve the optimal performance for each experiment separately.

**Numerical results:** The global accuracy is shown in Figure 4 under different settings. We can see that HyFEM algorithm can train a server model with higher accuracy than stand-alone training in all settings. Moreover, the server models can achieve comparable performance as models obtained with centralized training, even if none of the clients has full features or full classes of the data. HyFEM can deal with data with SH and FH. As expected, in setting ModelNet40:2, the server model accuracy is lower than in setting ModelNet40:1, because the matching problem is harder for larger classifiers and 12.08% of the data have never been used by any of the clients compared with centralized training.

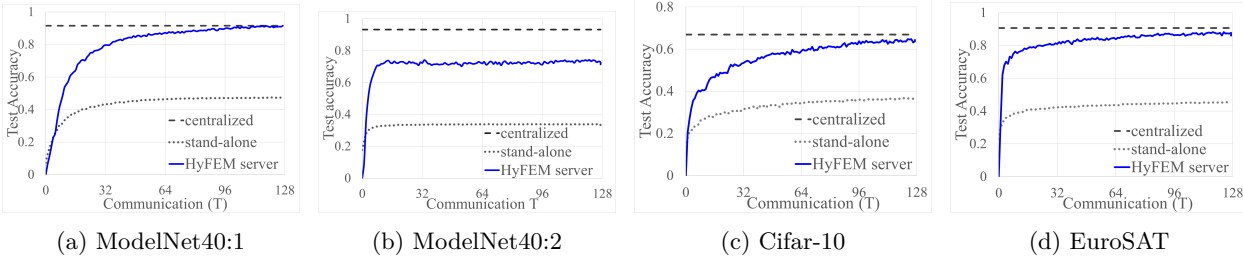

(a) ModelNet40:1      (b) ModelNet40:2      (c) Cifar-10      (d) EuroSAT

Figure 4: Test accuracy of server model trained with HyFEM compared with centralized training and stand-alone training for a) ModelNet40:1, b) ModelNet40:2, c) Cifar-10, and d) EuroSAT datasets.

The average client accuracy is shown in Figure 5 for different settings. Client models have lower testing accuracies compared with server models. This is reasonable as the client models are trained with partial features and biased data with partial classes. We also observe that the stand-alone accuracy under setting ModelNet40:1 is higher than ModelNet40:2, as each client has more samples. However, the accuracy improvement with HyFEM under setting ModelNet40:1 is less than setting ModelNet40:2, as the latter one uses more features. Nevertheless, HyFEM can train much better client models than stand-alone training even if the clients do not share the same input space and classes. By using the global model matching algorithm, the local classifiers can share knowledge with other clients on unseen classes and deal with SH.

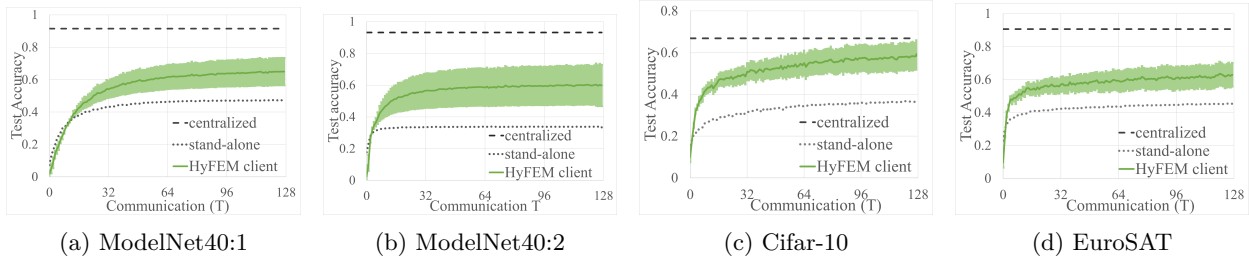

| (a) ModelNet40:1 | (b) ModelNet40:2 | (c) Cifar-10 | (d) EuroSAT |

Figure 5: Averaged test accuracy (with standard deviation) of all clients trained with HyFEM compared with centralized training and stand-alone training for a) ModelNet40:1, b) ModelNet40:2, c) Cifar-10, and d) EuroSAT datasets.

## 5 Conclusions

We propose a hybrid FL framework that handles a general collaborative-learning scenario with partially overlapped features and samples. We first clarify how the data are partitioned in the hybrid FL scenario and propose a generic problem formulation. Next, we show that the proposed formulation covers many horizontal and personalized FL settings, and develop a BCD-based algorithm, HyFEM, to solve the proposed problem. Finally, our numerical results on a number of image classification datasets demonstrate that HyFEM enables clients with partial features and samples to achieve a performance comparable to centralized training with full features.

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

# A Heterogeneous Model Matching Algorithm

In this section, we describe the details of the model-matching algorithm. First, we describe the motivation behind the design of the classifiers' regularizer (5), which encourages $w_m$'s to be matched together to construct $w_0$ is designed. Then we present the detailed version of line $8-14$ in Algorithm 2 for optimizing the regularizer (11).

## A.1 Regularizer Design

Recall the regularizer for the classifiers is given by:

$$r_{m,2}(w_m, \Pi_m, w_0) = \frac{1}{2} \|w_m - \Pi_m w_0\|^2, \quad \text{s.t. } \Pi_m \in \mathcal{S}(d_{m,w}, d_{0,w}),$$

where $d_{m,w}, d_{0,w}$ are the dimensions of $w_m, w_0$ and $\Pi_m$ is a selection matrix corresponding to the unknown pruning pattern. In this section, we explain why such a regularizer is used, and how the selection matrices $\Pi_m$'s are constructed.

We note that the proposed matching method is a non-trivial extension of the neural matching method (Yurochkin et al., 2019) (designed for horizontal FL) to the case of hybrid FL. In Yurochkin et al. (2019), the author considered the horizontal FL setting where the sizes and the functionalities of all the clients' models as well as the server's model are identical, so after matching, the clients can use the matched server-side model directly as their new model. However, in the considered hybrid FL setting, the input dimension of each of the client's inference block can be very different, and the server-side and client-side models do not share the same functionality. Therefore we cannot replace the client-side models with the server-side model. Such a special property of the hybrid FL problem introduces some significant challenges for the matching procedure. This is the main reason that in our proposed algorithm, the matching matrices and the client/server models have to be iteratively optimized.

Suppose that for each client $m$, its classifier $f_m(w_m; \cdot)$ has $L$ layers; then the inference block has the following structure:

$$y = \sigma_{m,L}(w_{m,L} \cdot \sigma_{m,L-1}(w_{m,L-1} \ldots \sigma_{m,1}(w_{m,1}\mathbf{v}_m) \ldots)), \tag{13}$$

where $\sigma_{m,l}(\cdot)$ represents the element-wise nonlinear activation function of layer $l$; $w_{m,l}$'s are the weight matrices of layer $l$, and $\mathbf{v}_m$ is the input of the classifier which is the stacked output of the feature extractors, i.e., $\mathbf{v}_m = H_m(\Theta_m; x)$. Then $w_m = \{w_{m,l}\}_{l=1}^{L}$; see Figure 6 for an illustration. Let us further define the server's output of feature extractors, activation functions and weights as $\mathbf{v}_0 \{\sigma_{0,l}(\cdots)\}_{l=1}^{L}$ similarly as above.

Note that giving a selection matrix with appropriate shape $\Pi \in \mathcal{S}$, left multiply the weight matrix $w_{m,l}$ by $\Pi^T$ ($\Pi^T w_{m,l}$) results in selecting the rows of $w_{m,l}$, which is equivalent to selecting the output neurons of the $l^{\text{th}}$ layer. And right multiply the weight matrix $w_{m,l}$ by $\Pi$ ($w_{m,l}\Pi$) results in selecting the columns of $w_{m,l}$, which is equivalent to selecting the input neurons of the $l^{\text{th}}$ layer.

The goal is to match $w_{m,l}$'s with the corresponding parameters $w_{0,l}$ at the server. Below, we discuss how the first layer, the middle layers, and the last layer are matched.

First, recall that the input of the classifiers has the following relation:

$$H_m(\Theta_m; x_m) = P_m H_0(\Theta_0; x_0), \text{that is,} \mathbf{v}_m = P_m \mathbf{v}_0,$$

where $P_m$ is the selection matrix defined by the feature overlapping pattern between $x_m$ and $x_0$. Then, let us multiply $P_m^T$ on both sides of the above equation, we obtain

$$P_m^T P_m \mathbf{v}_0 = P_m^T \mathbf{v}_m. \tag{14}$$

Note that $P_m^T \mathbf{v}_m$ pads zeros in the missing feature indices of the $\mathbf{v}_m$, so that it matches the size of $\mathbf{v}_0$. Let us define $\Pi_{m,1} = P_m$. By utilizing the fact that $P_m \in \mathcal{S}(d_m, d_0)$ is a selection matrix, it holds $P_m P_m^T = I_{d_m}$, then we have the following relation:

$$\sigma_{m,1}((w_{m,1}\Pi_{m,1})(\Pi_{m,1}^T \mathbf{v}_m)) = \sigma_{m,1}(w_{m,1}\mathbf{v}_m).$$

This process expands the input $\mathbf{v}_m$ to the same size as $\mathbf{v}_0$, while keeping the output of the first layer unchanged; see Fig. 7 for an illustration of this process.

Next, we would like to find a selection matrix $\Pi_{m,2} \in \mathcal{S}$ that compresses the output of the first layer of the server to match the output of the first layer of client $m$, as follows:

$$\sigma_{m,1}((w_{m,1}\Pi_{m,1})(\Pi_{m,1}^T \mathbf{v}_m)) \approx \Pi_{m,2} \cdot \sigma_{0,1}(w_{0,1}\mathbf{v}_0). \tag{15}$$

This output-matching relation imposes the following assumption on the model parameters:

$$w_{m,1} = \Pi_{m,2}w_{0,1}\Pi_{m,1}^T. \tag{16}$$

To see why (16) implies (15), we can plug (16) into the left hand side of (16), and obtain:

$$\begin{aligned}
\sigma_{m,1}((w_{m,1}\Pi_{m,1})(\Pi_{m,1}^T \mathbf{v}_m)) &= \sigma_{m,1}((\Pi_{m,2}w_{0,1}\Pi_{m,1}^T\Pi_{m,1})(\Pi_{m,1}^T \mathbf{v}_m)) \\
&\overset{(i)}{=} \Pi_{m,2} \cdot \sigma_{0,1}((w_{0,1}\Pi_{m,1}^T\Pi_{m,1})(\Pi_{m,1}^T \mathbf{v}_m)) \\
&= \Pi_{m,2} \cdot \sigma_{0,1}((w_{0,1}\Pi_{m,1}^T\Pi_{m,1})(\Pi_{m,1}^T\Pi_{m,1}\mathbf{v}_0)) \\
&= \Pi_{m,2} \cdot \sigma_{0,1}(w_{0,1}\Pi_{m,1}^T\Pi_{m,1}\mathbf{v}_0) \\
&\overset{(ii)}{\approx} \Pi_{m,2} \cdot \sigma_{0,1}(w_{0,1}\mathbf{v}_0),
\end{aligned} \tag{17}$$

where $(i)$ comes from the fact that projection only changes the order and pads zeros to the output, so applying element-wise activation before or after the projection does not affect the final output; in $(ii)$ we use the fact that $\Pi_{m,1}^T\Pi_{m,1}$ is a diagonal matrix with 1's and 0's on diagonal that can be approximated by a identity matrix. The above discussion suggests that, if (16) holds approximately, then (15) holds approximately. As a result, we design the regularizer on the first layer between client $m$ and the server, by approximately enforcing (16) as

$$\frac{1}{2}\left\|w_{m,1} - \Pi_{m,2}w_{0,1}\Pi_{m,1}^T\right\|^2.$$

Let us now analyze the constraint for $\Pi_{m,2}$. First, since the dimension of $w_{0,1}$ is larger or equal to that of $w_{m,1}$ for each client $m$, we require that each coordinate of $w_{m,1}$ is matched to one coordinate in $w_{0,1}$. Therefore we need $\Pi_{m,2}$ to satisfy $\mathbf{1} = \Pi_{m,2}\mathbf{1}$. Further, each coordinate in $w_{0,1}$ should match to a coordinate in at least one clients $m \in [M]$'s $w_{m,1}$, so this means $\sum_{m=1}^{M}\mathbf{1}^T\Pi_{m,2} \geq \mathbf{1}$. The above process is illustrated in Fig. 8.

For the $l^{\text{th}}$ middle layer, its input is the output of the previous layer. By fixing the projection matrices $\{\Pi_{m,l}\}_{m=1}^{M}$ that match the output of the $(l-1)^{\text{th}}$ layer at each client to the output of the $(l-1)^{\text{th}}$ layer at the server, the matching problem for the $l^{\text{th}}$ middle layer takes the same form as the matching problem for the input layer: the output of the previous layer $\sigma_{m,l-1}(\cdot)$ corresponds to the input $\mathbf{v}_m$; the projection matrices $\{\Pi_{m,l}\}_{m=1}^{M}$ correspond to the input projection matrices $\{\Pi_{m,1}\}_{m=1}^{M}$; the goal is to find the projection matrices $\{\Pi_{m,l+1}\}_{m=1}^{M}$ that match the output of the $l^{\text{th}}$ layer at each client to the output of the same layer as the server.

By using the same argument that we used for the input layer, we design the regularizer on the $l^{\text{th}}$ middle layer between client $m$ and the server as

$$\frac{1}{2}\left\|w_{m,l} - \Pi_{m,l+1}w_{0,l}\Pi_{m,l}^T\right\|^2.$$

For the last layer, the output at each client is the same as the server, which is the predicted label. Therefore, the projection matrix of the output for the last layer is an identity matrix, and we design the regularizer for the last layer as

$$\frac{1}{2}\left\|w_{m,L} - w_{0,L}\Pi_{m,L}^T\right\|^2.$$

Next, by vectorizing $w_{m,l}$'s we have following relation:

$$\text{vec}(\Pi_{m,l+1}w_{0,l}\Pi_{m,l}^T) = (\Pi_{m,l} \otimes \Pi_{m,l+1})\text{vec}(w_{0,l}),$$

---

**Algorithm 2** Sub-routines for Algorithm 1

1: **ClientUpdate**$(\Theta_0^t, \Pi_m^t, w_0^t, Q, \eta)$
2: **Initialize:** $\Theta_m^{t,0} \leftarrow P_m \Theta_0^t, w_m^{t,0} \leftarrow \Pi_m^t w_0^t$
3: **for** $q = 0, \ldots, Q-1$ **do**
4: $\quad$ Uniformly sample $n \in \mathcal{N}_m$
5: $\quad \Theta_m^{t,q+1} \leftarrow \Theta_m^{t,q} - \eta \left( \nabla_{\Theta_m} \ell(F_m(w_m^{t,q}; H_m(\Theta_m^{t,q}; x_{m,n})), y_n) + \mu_1(\Theta_m^{t,q} - P_m \Theta_0^t) \right)$
6: $\quad w_m^{t,q+1} \leftarrow w_m^{t,q} - \eta \left( \nabla_{w_m} \ell(F_m(w_m^{t,q}; H_m(\Theta_m^{t,q}; x_{m,n})), y_n) + \mu_2(w_m^{t,q} - \Pi_m^t w_0^t) \right)$
7: **Output:** $\Theta_m^{t,Q}, w_m^{t,Q}$
8: **ModelMatching**$(\{w_m^{t,Q}, \Pi_m^t\}_{m=1}^M, w_0^t, P)$
9: **for** $p = 0, \ldots, P-1$ **do**
10: $\quad$ **for** $m' = 1, \ldots, M$ in parallel **do**
11: $\qquad \hat{w}_0^{t,p} \leftarrow \left( \sum_{m \neq m'} p_m (\Pi_m^{t,p})^T \Pi_m^{t,p} \right)^{-1} \left( \sum_{m \neq m'} p_m (\Pi_m^{t,p})^T w_m^{t,Q} \right)$
12: $\qquad \Pi_{m'}^{t,p+1} \leftarrow \arg\min_{\Pi_{m'}} r_{m',2}(w_{m'}^{t,Q}, \Pi_{m'}, \hat{w}_0^{t,p}),$ $\quad$ // Using Hungarian algorithm
13: $w_0^{t+1} \leftarrow \left( \sum_{m=1}^M p_m (\Pi_m^{t,P})^T \Pi_m^{t,P} \right)^{-1} \left( \sum_{m=1}^M p_m (\Pi_m^{t,P})^T w_m^{t,Q} \right), \Pi_m^{t+1} \leftarrow \Pi_m^{t,P}$
14: **Output:** $w_0^{t+1}, \{\Pi_m^{t+1}\}_{m=1}^M$

---

where $\otimes$ denotes the Kronecker product. Therefore the regularizer for each layer can be rewritten as

$$\frac{1}{2} \left\| \text{vec}(w_{m,l}) - (\Pi_{m,l} \otimes \Pi_{m,l+1}) \text{vec}(w_{0,l}) \right\|^2.$$

Finally we can stack the sub-vector $\{\text{vec}(w_{m,l})\}_{l=1}^L$ into $w_m$, and define the projection matrix of the long vector as

$$\Pi_m := \text{diag}((\Pi_{m,1} \otimes \Pi_{m,2}), \ldots, (\Pi_{m,L} \otimes \mathbf{I})).$$

Again, $\Pi_m$ is a block diagonal matrix, and it is easy to verify that it satisfies the following conditions:

$$\Pi_m^T \mathbf{1} = \mathbf{1}, \ \sum_{m=1}^M \mathbf{1}\Pi_m \geq \mathbf{1}; \Pi_m \geq \mathbf{0}. \tag{18}$$

Finally, we obtain the final formulation of the regularizer

$$r_{m,2}(w_m, \Pi_m w_0) = \frac{1}{2} \left\| w_m - \Pi_m w_0 \right\|^2, \ \forall \ m \in [M]$$

$$\text{where} \ \ \Pi_m\text{'s} \ \ \text{satisfy:} \ \Pi_m \in \mathcal{S}(d_{w,m}, d_{w,0}), \ \sum_{m=1}^M \mathbf{1}_{d_{w,m}}^T \Pi_m \geq \mathbf{1}_{d_{w,0}}^T. \tag{19}$$

## A.2 Optimization Procedure

In this subsection, we describe the detailed procedures in line $8-14$ of Algorithm 2 to optimize the classifier matching problem (5) or its more detailed formulation (19). The procedure with more details is given in Algorithm 3.

We iteratively solve the matching problem (11) for $P$ iterations. In each iteration, we randomly pick a client $m'$ to match it with the server's classifier from the first layer to the last layer.

For the first $L-1$ layers, we first fix $\Pi_{m',l-1}$ and construct an assignment cost matrix $C_l$ that computes the cost to match $j^{\text{th}}$ row in client $m'$ to $i^{\text{th}}$ row in the server of layer $l$ for all $(i, j)$. The element $C_l(i, j)$ of the cost matrix is defined as:

$$C_l(i, j) = \begin{cases} \text{dist}_1(w_{0,l} \Pi_{m,l-1}^T[i], w_{m,l}[j]) & w_{0,l} \Pi_{m,l-1}^T[i] \neq \mathbf{0} \\ \text{dist}_2(w_{m,l}[j]) & \text{otherwise,} \end{cases} \tag{20}$$

where $w_{0,l} \Pi_{m,l-1}^T[i]$ and $w_{m,l}[j]$ denote the $i^{\text{th}}$ and $j^{\text{th}}$ row of the matrices, $\text{dist}_1$ is the similarity cost for matching $w_{m,l}[j]$ to an existing row, and $\text{dist}_2$ is the dimension penalty to match $w_{m,l}[j]$ to a new row in

---

**Algorithm 3** Model Matching Procedure

---

1: **ModelMatching**
2: **Input:** $\{w_m^{t,Q}, \Pi_m^t\}_{m=1}^M, w_0^t, P$
3: **for** $p = 0, \ldots, P - 1$ **do**
4:      Uniformly sample $m' \in [M]$
5:      $w_0^{t,p} \leftarrow \left( \sum_{m \neq m'} p_m (\Pi_m^{t,p})^T \Pi_m^{t,p} \right)^{-1} \left( \sum_{m \neq m'} p_m (\Pi_m^{t,p})^T w_m^{t,Q} \right)$
6:      **for** $l = 1, \ldots, L - 1$: **do**
7:           Construct cost matrix $C_l$ with (20).
8:           $\Pi_{m',l}^{t,p+1} \leftarrow \arg\min_{\Pi_{m'}} \sum_{i,j} \Pi_{m'}(i,j) \cdot C_l(i,j),$      *// Using Hungarian algorithm*
9: $w^{t+1} \leftarrow \left( \sum_{m=1}^M p_m (\Pi_m^{t,P})^T \Pi_m^{t,P} \right)^{-1} \left( \sum_{m=1}^M p_m (\Pi_m^{t,P})^T w_m^{t,Q} \right), \Pi_m^{t+1} \leftarrow \Pi_m^{t,P}$
10: **Output:** $w_0^{t+1}, \{\Pi_m^{t+1}\}_{m=1}^M$

---

$w_{0,l}$. One specification of the cost functions is PFNM (Yurochkin et al., 2019) that uses the MAP loss of the Beta-Bernoulli process, where $\text{dist}_1$ is based on the Gaussian prior and $\text{dist}_2$ follows the Indian Buffet process prior. Then we can solve the assignment problem to obtain $\Pi_{m',l}$ with the celebrated Hungarian algorithm (Kuhn, 1955).

Note that for the first layer, the matching pattern $\Pi_{m,0}$ is given by $\Pi_{m,0} = P_m$. And we do not need to match the output layer.

## B    Convergence Analysis

In this section, we analyze the convergence property of Algorithm 1. We first make the following assumptions on the problem:

**A 1 (Block Lipschitz Gradient)** *For each parameter blocks in $\{w_m, \Theta_m\}_{m=1}^M$, there exists an $L_m$ such that the following holds:*

$$\left\| \nabla_{\Theta_m} f_m(\Theta_m, w_m) - \nabla_{\Theta'_m} f_m(\Theta'_m, w'_m) \right\| + \left\| \nabla_{w_m} f_m(\Theta_m, w_m) - \nabla_{w'_m} f_m(\Theta'_m, w'_m) \right\|$$
$$\leq L_m \left( \left\| \Theta_m - \Theta'_m \right\| + \left\| w_m - w'_m \right\| \right), \qquad \forall\, \Theta_m, \Theta'_m, w_m, w'_m,$$

**A 2 (Lower Bounded Loss)** *There exist finite lower bounds for each client classification loss, i.e.,*

$$\exists\, \underline{f_m} > -\infty, \quad \text{s.t. } f_m(\Theta_m, w_m) \geq \underline{f_m}, \quad \forall \Theta_m, w_m, m.$$

**A 3 (Bounded Variance)** *The stochastic partial gradient estimation has bounded variance $\sigma_\Theta^2$ and $\sigma_w^2$, i.e.,*

$$\mathbb{E}_n \left\| \nabla_{\Theta_m} \ell(F_m(w_m; H_m(\Theta_m; x_{m,n})), y_n) - \nabla_{\Theta_m} f_m(w_m, \Theta_m) \right\|^2 \leq \sigma_\Theta^2, \forall\, \Theta_m, w_m, \ \forall\, m \in [M],$$
$$\mathbb{E}_n \left\| \nabla_{w_m} \ell(F_m(w_m; H_m(\Theta_m; x_{m,n})), y_n) - \nabla_{w_m} f_m(w_m, \Theta_m) \right\|^2 \leq \sigma_w^2, \forall\, \Theta_m, w_m, \ \forall\, m \in [M].$$

We can abstract HyFEM to a BCD-type algorithm by redefining the model parameters and the problem as follows:

1. Define $\mathbf{x} := [\Theta_1; \ldots; \Theta_M; w_1; \ldots; w_M]$, $\mathbf{y} := [\text{vec}(\Pi_1); \ldots, \text{vec}(\Pi_M)]$, and $\mathbf{z} := [\Theta_0; w_0]$.

2. Define $\mathcal{L}(\mathbf{x}, \mathbf{y}, \mathbf{z}) := \sum_{m=1}^M p_m \left( f_m(\Theta_m, w_m) + \mu_1 \cdot r_{m,1}(\Theta_m, \Theta_0) + \mu_2 \cdot r_{m,2}(w_m, \Pi_m, w_0) \right)$.

Then the optimization problem (6) can be simplified as:

$$\min_{\mathbf{x}, \mathbf{y}, \mathbf{z}} \mathcal{L}(\mathbf{x}, \mathbf{y}, \mathbf{z}), \quad \text{s.t.} \quad \mathbf{y} \in \text{Range}(\mathbf{y}). \tag{21}$$

Moreover, the algorithm can be simplified as:

$$\mathbf{x}^{t,q+1} = \mathbf{x}^{t,q} - \eta \tilde{\nabla}_{\mathbf{x}} \mathcal{L}(\mathbf{x}^{t,q}, \mathbf{y}^t, \mathbf{z}^t), \text{ for } q = 0, \dots, Q-1 \tag{22a}$$

$$\mathbf{y}^{t+1} = \underset{\mathbf{y} \in \text{Range}(\mathbf{y})}{\arg\min} \mathcal{L}(\mathbf{x}^{t,Q}, \mathbf{y}, \mathbf{z}^t), \tag{22b}$$

$$\mathbf{z}^{t+1} = \underset{\mathbf{z}}{\arg\min} \mathcal{L}(\mathbf{x}^{t,Q}, \mathbf{y}^{t+1}, \mathbf{z}), \tag{22c}$$

where we assume $\mathbf{x}^{t+1,0} = \mathbf{x}^{t+1} = \mathbf{x}^{t,Q}$ and $\tilde{\nabla}_{\mathbf{x}} \mathcal{L}(\cdot)$ denotes the stochastic partial gradient of $\mathbf{x}$.

We make the following assumptions to problem (21).

**A 4 (Block Lipschitz Gradient)** *$\mathcal{L}$ is block smooth, and for parameter $\mathbf{x}$ and $\mathbf{z}$, there exists positive constants $L_{\mathbf{x}}, L_{\mathbf{z}}$ and $C_{\mathbf{x}}$ such that the following holds:*

$$\|\nabla_{\mathbf{x}} \mathcal{L}(\mathbf{x}, \mathbf{y}, \mathbf{z}) - \nabla_{\mathbf{x}'} \mathcal{L}(\mathbf{x}', \mathbf{y}, \mathbf{z})\| \leq L_{\mathbf{x}} \|\mathbf{x} - \mathbf{x}'\|, \forall \, \mathbf{x}, \mathbf{x}', \mathbf{z}, \forall \mathbf{y} \in \text{Range}(\mathbf{y}).$$
$$\|\nabla_{\mathbf{x}} \mathcal{L}(\mathbf{x}, \mathbf{y}, \mathbf{z}) - \nabla_{\mathbf{x}} \mathcal{L}(\mathbf{x}, \mathbf{y}, \mathbf{z}')\| \leq C_{\mathbf{x}} \|\mathbf{z} - \mathbf{z}'\|, \forall \, \mathbf{z}, \mathbf{z}', \mathbf{x}, \forall \mathbf{y} \in \text{Range}(\mathbf{y}).$$

**A 5 (Block Strong Convexity of z)** *For parameter $\mathbf{z}$, there exists a positive constant $\mu$ such that the following holds:*

$$\mathcal{L}(\mathbf{x}, \mathbf{y}, \mathbf{z}') \geq \mathcal{L}(\mathbf{x}, \mathbf{y}, \mathbf{z}) + \langle \nabla_{\mathbf{z}} \mathcal{L}(\mathbf{x}, \mathbf{y}, \mathbf{z}), \mathbf{z}' - \mathbf{z} \rangle + \frac{\mu}{2} \|\mathbf{z}' - \mathbf{z}\|^2, \forall \, \mathbf{x}, \mathbf{z}, \mathbf{z}', \forall \mathbf{y} \in \text{Range}(\mathbf{y}).$$

**A 6 (Unbiased Stochastic Partial Gradient)** *The stochastic partial gradient of $\mathbf{x}$ is unbiased:*

$$\mathbb{E} \, \tilde{\nabla}_{\mathbf{x}} \mathcal{L}(\mathbf{x}, \mathbf{y}, \mathbf{z}) = \nabla_{\mathbf{x}} \mathcal{L}(\mathbf{x}, \mathbf{y}, \mathbf{z}), \forall \, \mathbf{x}, \mathbf{z}, \forall \mathbf{y} \in \text{Range}(\mathbf{y}).$$

**A 7 (Bounded Variance of Stochastic Partial Gradient)** *The stochastic partial gradient of $\mathbf{x}$ has bounded variance $\sigma^2$:*

$$\mathbb{E} \left\| \tilde{\nabla}_{\mathbf{x}} \mathcal{L}(\mathbf{x}, \mathbf{y}, \mathbf{z}) - \nabla_{\mathbf{x}} \mathcal{L}(\mathbf{x}, \mathbf{y}, \mathbf{z}) \right\|^2 \leq \sigma^2, \forall \, \mathbf{x}, \mathbf{z}, \forall \mathbf{y} \in \text{Range}(\mathbf{y}).$$

**A 8 (Lower Bounded Function)** *The problem $\mathcal{L}$ is bounded from below, i.e.,*

$$\exists \underline{\mathcal{L}} > -\infty, \text{s.t. } \mathcal{L}(\mathbf{x}, \mathbf{y}, \mathbf{z}) \geq \underline{\mathcal{L}}, \quad \forall \mathbf{x}, \mathbf{z}, \forall \mathbf{y} \in \text{Range}(\mathbf{y}).$$

**A 9 (Compact Constraint Set)** *For parameter $\mathbf{y}$, the constraint set $Range(\mathbf{y})$ is compact.*

Note that in A4 and A5, we only assume blocks $\mathbf{x}, \mathbf{z}$ are smooth, and only block $\mathbf{z}$ is strongly convex while block $\mathbf{y}$ can be non-smooth and non-convex and $\mathbf{x}$ can potentially be non-convex. Further we assume that $\nabla_{\mathbf{x}} \mathcal{L}$ is smooth w.r.t. $\mathbf{z}$, which is non-standard, but we can prove that it holds for problem (6). The remaining assumptions A6-A9 are common when analyzing stochastic algorithms. Further, we can verify that the above assumptions hold for the original problem (6).

**Lemma 1** *Suppose (6) satisfies assumptions A1-A3, then it satisfies A4-A9 with the constants in the assumptions given as:*

$$L_{\mathbf{x}} = \max_{m}\{p_m L_m + \max\{\mu_1, \mu_2\}\}, \; C_{\mathbf{x}} = \max\{\mu_1, \mu_2\},$$

$$\mu \geq \min_{m}\{p_m\} \cdot \min\{\mu_1, \mu_2\}, \; \sigma^2 = \sigma_\Theta^2 + \sigma_w^2, \; \underline{\mathcal{L}} = \sum_{m=1}^{M} p_m \underline{f_m}.$$

The proof is given in Section B.2.

Then we have the following result:

**Theorem 2** *Suppose the problem* (21) *satisfies A4-A9 and run* (22) *for $T$ iterations with stepsize $\eta \le \min\{\frac{1}{L_{\mathbf{x}}}, \frac{8\mu}{5C_{\mathbf{x}}^2}\}$. Then the sequence $\{\mathbf{x}^{t,q}, \mathbf{y}^t, \mathbf{z}^t\}_{t=0}^T$ generated by* (22) *satisfies:*

$$\frac{1}{TQ}\sum_{t=0}^{T-1}\mathbb{E}\left(\frac{\mu}{\eta}\left\|\mathbf{z}^{t+1}-\mathbf{z}^t\right\|^2 + \left\|\nabla_{\mathbf{x}}\mathcal{L}(\mathbf{x}^{t+1}, \mathbf{y}^{t+1}, \mathbf{z}^t)\right\|^2 + \sum_{q=0}^{Q}\left\|\nabla_{\mathbf{x}}\mathcal{L}(\mathbf{x}^{t,q}, \mathbf{y}^t, \mathbf{z}^t)\right\|^2\right)$$

$$\le \frac{10}{TQ\eta}\left(\mathcal{L}(\mathbf{x}^0, \mathbf{y}^0, \mathbf{z}^0) - \underline{\mathcal{L}}\right) + \left(5L_{\mathbf{x}}\eta + \frac{2L_{\mathbf{x}}^2\eta^2}{Q}\right)\sigma^2, \tag{23}$$

*and* $\left\|\nabla_{\mathbf{z}}\mathcal{L}(\mathbf{x}^t, \mathbf{y}^t, \mathbf{z}^t)\right\|^2 = 0, \ \forall t \in [T].$

This result indicates that by setting $Q = T, \eta = \sqrt{\frac{2(\mathcal{L}(\mathbf{x}^0, \mathbf{y}^0, \mathbf{z}^0) - \underline{\mathcal{L}})}{L_{\mathbf{x}}QT\sigma^2}}$, the right-hand-side (RHS) of (23) becomes $\frac{10\sigma\sqrt{2(\mathcal{L}(\mathbf{x}^0, \mathbf{y}^0, \mathbf{z}^0) - \underline{\mathcal{L}})L_{\mathbf{x}}}}{T} + \frac{2L_{\mathbf{x}}(\mathcal{L}(\mathbf{x}^0, \mathbf{y}^0, \mathbf{z}^0) - \underline{\mathcal{L}})}{T^3} = \mathcal{O}(\frac{1}{T})$. Let us analyze the left-hand-side (LHS) terms of (23). First, we have

$$\frac{1}{TQ}\sum_{t=0}^{T-1}\mathbb{E}\frac{\mu}{\eta}\left\|\mathbf{z}^{t+1}-\mathbf{z}^t\right\|^2 = \frac{\mu}{T}\sum_{t=0}^{T-1}\mathbb{E}\left\|\mathbf{z}^{t+1}-\mathbf{z}^t\right\|^2 = \mathcal{O}\left(\frac{1}{T}\right),$$

indicating that $\mathbb{E}\left\|\mathbf{z}^{t+1}-\mathbf{z}^t\right\|^2 = \mathcal{O}\left(\frac{1}{T}\right)$. Second, we have

$$\frac{1}{TQ}\sum_{t=0}^{T-1}\mathbb{E}\left(\left\|\nabla_{\mathbf{x}}\mathcal{L}(\mathbf{x}^{t+1}, \mathbf{y}^{t+1}, \mathbf{z}^t)\right\|^2 + \sum_{q=0}^{Q}\left\|\nabla_{\mathbf{x}}\mathcal{L}(\mathbf{x}^{t,q}, \mathbf{y}^t, \mathbf{z}^t)\right\|^2\right) = \mathcal{O}\left(\frac{1}{T}\right),$$

where the LHS is the sum of $T(Q + 2)$ terms of $\|\nabla_{\mathbf{x}}\mathcal{L}\|^2$ divide by $TQ$, which also indicates that $\mathbb{E}\left\|\nabla_{\mathbf{x}}\mathcal{L}(\mathbf{x}^{t,q}, \mathbf{y}^t, \mathbf{z}^t)\right\|^2 = \mathcal{O}\left(\frac{1}{T}\right)$. Together we have that algorithm (22) finds a stationary solution of (21) w.r.t. $\mathbf{x}, \mathbf{z}$ with rate $\mathcal{O}\left(\frac{1}{T}\right)$. Combining Theorem 2 with Lemma 1, we have that by running Algorithm 1, parameters $\{w_m, \Theta_m\}_{m=0}^M$ converges to their stationary point of (6), while $\{\Pi_m\}_{m=1}^M$ stays in a compact set.

Alternatively, if we assume the solution to $\mathbf{y}$ is unique, and update on $\mathbf{x}$ is a one-step gradient descent, i.e., $Q = 1$ and

$$\mathbf{x}^{t+1} = \mathbf{x}^t - \eta\nabla_{\mathbf{x}}\mathcal{L}(\mathbf{x}^t, \mathbf{y}^t, \mathbf{z}^t),$$

then by applying (Razaviyayn et al., 2013, Theorem 2), Algorithm 1 asymptotically converges to the first-order stationary point of (6).

### B.1 Proof for Theorem 2

We begin by proving the following descent result:

$$\mathbb{E}^t\,\mathcal{L}(\mathbf{x}^{t+1}, \mathbf{y}^{t+1}, \mathbf{z}^{t+1}) - \mathcal{L}(\mathbf{x}^t, \mathbf{y}^t, \mathbf{z}^t) \le -\frac{\eta}{2}\sum_{q=0}^{Q-1}\mathbb{E}^t\left\|\nabla_{\mathbf{x}}\mathcal{L}(\mathbf{x}^{t,q}, \mathbf{y}^t, \mathbf{z}^t)\right\|^2$$

$$-\frac{\mu}{2}\left\|\mathbf{z}^{t+1}-\mathbf{z}^t\right\|^2 + \frac{QL_{\mathbf{x}}\eta^2\sigma^2}{2}, \tag{24}$$

where we denote the expectation conditioned on the information up to iteration $t$ as $\mathbb{E}^t$. First, we write the LHS of the above equation into three terms as below:

$$\mathcal{L}(\mathbf{x}^{t+1}, \mathbf{y}^{t+1}, \mathbf{z}^{t+1}) - \mathcal{L}(\mathbf{x}^t, \mathbf{y}^t, \mathbf{z}^t) = \left(\mathcal{L}(\mathbf{x}^{t+1}, \mathbf{y}^{t+1}, \mathbf{z}^{t+1}) - \mathcal{L}(\mathbf{x}^{t+1}, \mathbf{y}^{t+1}, \mathbf{z}^t)\right)$$

$$+ \left(\mathcal{L}(\mathbf{x}^{t+1}, \mathbf{y}^{t+1}, \mathbf{z}^t) - \mathcal{L}(\mathbf{x}^{t+1}, \mathbf{y}^t, \mathbf{z}^t)\right) \tag{25}$$

$$+ \left(\mathcal{L}(\mathbf{x}^{t+1}, \mathbf{y}^t, \mathbf{z}^t) - \mathcal{L}(\mathbf{x}^t, \mathbf{y}^t, \mathbf{z}^t)\right).$$

We bound the three terms on the RHS of the above equation separately.

**1)** The first term $\left(\mathcal{L}(\mathbf{x}^{t+1}, \mathbf{y}^{t+1}, \mathbf{z}^{t+1}) - \mathcal{L}(\mathbf{x}^{t+1}, \mathbf{y}^{t+1}, \mathbf{z}^t)\right)$ can be bounded by applying A5:

$$
\begin{aligned}
\mathcal{L}(\mathbf{x}^{t+1}, & \mathbf{y}^{t+1}, \mathbf{z}^{t+1}) - \mathcal{L}(\mathbf{x}^{t+1}, \mathbf{y}^{t+1}, \mathbf{z}^t) \\
& \stackrel{A5}{\leq} -\left\langle \nabla \mathcal{L}_{\mathbf{z}}(\mathbf{x}^{t+1}, \mathbf{y}^{t+1}, \mathbf{z}^{t+1}), \mathbf{z}^t - \mathbf{z}^{t+1} \right\rangle - \frac{\mu}{2} \left\| \mathbf{z}^{t+1} - \mathbf{z}^t \right\|^2 \\
& \stackrel{(a)}{=} -\frac{\mu}{2} \left\| \mathbf{z}^{t+1} - \mathbf{z}^t \right\|^2,
\end{aligned}
\tag{26}
$$

where in $(a)$ uses update rule (22c) that by exact minimization $\nabla_{\mathbf{z}} \mathcal{L}(\mathbf{x}^{t+1}, \mathbf{y}^{t+1}, \mathbf{z}^{t+1}) = 0$.

**2)** By the update rule (22b), the second term $\left(\mathcal{L}(\mathbf{x}^{t+1}, \mathbf{y}^{t+1}, \mathbf{z}^t) - \mathcal{L}(\mathbf{x}^{t+1}, \mathbf{y}^t, \mathbf{z}^t)\right)$ can be bound by

$$
\mathcal{L}(\mathbf{x}^{t+1}, \mathbf{y}^{t+1}, \mathbf{z}^t) - \mathcal{L}(\mathbf{x}^{t+1}, \mathbf{y}^t, \mathbf{z}^t) \leq 0.
\tag{27}
$$

**3)** The third term $\left(\mathcal{L}(\mathbf{x}^{t+1}, \mathbf{y}^t, \mathbf{z}^t) - \mathcal{L}(\mathbf{x}^t, \mathbf{y}^t, \mathbf{z}^t)\right)$ can be further decompose into:

$$
\begin{aligned}
\mathcal{L}(\mathbf{x}^{t+1}, \mathbf{y}^t, \mathbf{z}^t) - \mathcal{L}(\mathbf{x}^t, \mathbf{y}^t, \mathbf{z}^t) &= \mathcal{L}(\mathbf{x}^{t,Q}, \mathbf{y}^t, \mathbf{z}^t) - \mathcal{L}(\mathbf{x}^{t,0}, \mathbf{y}^t, \mathbf{z}^t) \\
&= \sum_{q=0}^{Q-1} \left( \mathcal{L}(\mathbf{x}^{t,q+1}, \mathbf{y}^t, \mathbf{z}^t) - \mathcal{L}(\mathbf{x}^{t,q}, \mathbf{y}^t, \mathbf{z}^t) \right),
\end{aligned}
\tag{28}
$$

where the first inequality uses the definition that $\mathbf{x}^{r,Q} = \mathbf{x}^{r+1}$ and $\mathbf{x}^{r,0} = \mathbf{x}^r$. Then we bound each term in the summation as:

$$
\begin{aligned}
\mathcal{L}(\mathbf{x}^{t,q+1}, \mathbf{y}^t, \mathbf{z}^t) - \mathcal{L}(\mathbf{x}^{t,q}, \mathbf{y}^t, \mathbf{z}^t) &\stackrel{A4}{\leq} \left\langle \nabla_{\mathbf{x}} \mathcal{L}(\mathbf{x}^{t,q}, \mathbf{y}^t, \mathbf{z}^t), \mathbf{x}^{r,q+1} - \mathbf{x}^{r,q} \right\rangle + \frac{L_{\mathbf{x}}}{2} \left\| \mathbf{x}^{t,q+1} - \mathbf{x}^{t,q} \right\|^2 \\
&\stackrel{(22a)}{=} -\eta \left\langle \nabla_{\mathbf{x}} \mathcal{L}(\mathbf{x}^{t,q}, \mathbf{y}^t, \mathbf{z}^t), \tilde{\nabla}_{\mathbf{x}} \mathcal{L}(\mathbf{x}^{t,q}, \mathbf{y}^t, \mathbf{z}^t) \right\rangle + \frac{L_{\mathbf{x}} \eta^2}{2} \left\| \tilde{\nabla}_{\mathbf{x}} \mathcal{L}(\mathbf{x}^{t,q}, \mathbf{y}^t, \mathbf{z}^t) \right\|^2.
\end{aligned}
\tag{29}
$$

Taking expectation on $(t, q)$, we have:

$$
\begin{aligned}
\mathbb{E}^{t,q} \mathcal{L}(\mathbf{x}^{t,q+1}, & \mathbf{y}^t, \mathbf{z}^t) - \mathcal{L}(\mathbf{x}^{t,q}, \mathbf{y}^t, \mathbf{z}^t) \\
& \leq -\eta \left\langle \nabla_{\mathbf{x}} \mathcal{L}(\mathbf{x}^{t,q}, \mathbf{y}^t, \mathbf{z}^t), \mathbb{E}^{t,q} \tilde{\nabla}_{\mathbf{x}} \mathcal{L}(\mathbf{x}^{t,q}, \mathbf{y}^t, \mathbf{z}^t) \right\rangle + \frac{L_{\mathbf{x}} \eta^2}{2} \mathbb{E}^{t,q} \left\| \tilde{\nabla}_{\mathbf{x}} \mathcal{L}(\mathbf{x}^{t,q}, \mathbf{y}^t, \mathbf{z}^t) \right\|^2 \\
& \stackrel{(a)}{=} \eta \left\| \nabla_{\mathbf{x}} \mathcal{L}(\mathbf{x}^{t,q}, \mathbf{y}^t, \mathbf{z}^t) \right\|^2 + \frac{L_{\mathbf{x}} \eta^2}{2} \left\| \nabla_{\mathbf{x}} \mathcal{L}(\mathbf{x}^{t,q}, \mathbf{y}^t, \mathbf{z}^t) \right\|^2 \\
& \quad + \frac{L_{\mathbf{x}} \eta^2}{2} \mathbb{E}^{t,q} \left\| \tilde{\nabla}_{\mathbf{x}} \mathcal{L}(\mathbf{x}^{t,q}, \mathbf{y}^t, \mathbf{z}^t) - \nabla_{\mathbf{x}} \mathcal{L}(\mathbf{x}^{t,q}, \mathbf{y}^t, \mathbf{z}^t) \right\|^2 \\
& \stackrel{A7}{\leq} -\left( \eta - \frac{L_{\mathbf{x}} \eta^2}{2} \right) \left\| \nabla_{\mathbf{x}} \mathcal{L}(\mathbf{x}^{t,q}, \mathbf{y}^t, \mathbf{z}^t) \right\|^2 + \frac{L_{\mathbf{x}} \eta^2 \sigma^2}{2},
\end{aligned}
\tag{30}
$$

where $(a)$ first applies the fact that $\mathbb{E}(X^2) = (\mathbb{E} X)^2 + \mathbb{E}((X - E(X))^2)$ to the second therm, then applies A6 to the first and the second term.

By picking $\eta \leq \frac{1}{L_{\mathbf{x}}}$ and substituting (30) to (28), we have:

$$
\mathbb{E}^t \mathcal{L}(\mathbf{x}^{t+1}, \mathbf{y}^t, \mathbf{z}^t) - \mathcal{L}(\mathbf{x}^t, \mathbf{y}^t, \mathbf{z}^t) \leq -\frac{\eta}{2} \sum_{q=0}^{Q-1} \mathbb{E}^t \left\| \nabla_{\mathbf{x}} \mathcal{L}(\mathbf{x}^{t,q}, \mathbf{y}^t, \mathbf{z}^t) \right\|^2 + \frac{Q L_{\mathbf{x}} \eta^2 \sigma^2}{2}.
\tag{31}
$$

Then we substitute (26), (27) and (31) back to (25), then we obtain (24).

To prove Theorem 2, we need to further bound $\left\|\nabla_{\mathbf{x}}\mathcal{L}(\mathbf{x}^{t+1},\mathbf{y}^t,\mathbf{z}^t)\right\|^2$ and $\left\|\nabla_{\mathbf{x}}\mathcal{L}(\mathbf{x}^{t+1},\mathbf{y}^{t+1},\mathbf{z}^t)\right\|^2$. We bound them as follows. Term $\left\|\nabla_{\mathbf{x}}\mathcal{L}(\mathbf{x}^{t+1},\mathbf{y}^t,\mathbf{z}^t)\right\|^2$ can be bound as:

$$
\begin{aligned}
\mathbb{E}^{t,Q-1}\left\|\nabla_{\mathbf{x}}\mathcal{L}(\mathbf{x}^{t+1},\mathbf{y}^t,\mathbf{z}^t)\right\|^2 &\overset{(a)}{\leq} 2\left\|\nabla_{\mathbf{x}}\mathcal{L}(\mathbf{x}^{t,Q-1},\mathbf{y}^t,\mathbf{z}^t)\right\|^2 \\
&\quad + \mathbb{E}^{t,Q-1}\,2\left\|\nabla_{\mathbf{x}}\mathcal{L}(\mathbf{x}^{t,Q},\mathbf{y}^t,\mathbf{z}^t) - \nabla_{\mathbf{x}}\mathcal{L}(\mathbf{x}^{t,Q-1},\mathbf{y}^t,\mathbf{z}^t)\right\|^2 \\
&\overset{A4}{\leq} 2L_{\mathbf{x}}^2\,\mathbb{E}^{t,Q-1}\left\|\mathbf{x}^{t,Q} - \mathbf{x}^{t,Q-1}\right\|^2 + 2\left\|\nabla_{\mathbf{x}}\mathcal{L}(\mathbf{x}^{t,Q-1},\mathbf{y}^t,\mathbf{z}^t)\right\|^2 \\
&\overset{(22a)}{=} 2L_{\mathbf{x}}^2\eta^2\,\mathbb{E}^{t,Q-1}\left\|\tilde{\nabla}_{\mathbf{x}}\mathcal{L}(\mathbf{x}^{t,Q-1},\mathbf{y}^t,\mathbf{z}^t)\right\|^2 + 2\left\|\nabla_{\mathbf{x}}\mathcal{L}(\mathbf{x}^{t,Q-1},\mathbf{y}^t,\mathbf{z}^t)\right\|^2 \\
&\overset{A7}{\leq} \left(2+2L_{\mathbf{x}}^2\eta^2\right)\left\|\nabla_{\mathbf{x}}\mathcal{L}(\mathbf{x}^{t,Q-1},\mathbf{y}^t,\mathbf{z}^t)\right\|^2 + 2L_{\mathbf{x}}^2\eta^2\sigma^2,
\end{aligned}
\tag{32}
$$

where in $(a)$ we add and subtract $\nabla_{\mathbf{x}}\mathcal{L}(\mathbf{x}^{t,Q-1},\mathbf{y}^t,\mathbf{z}^t)$ and apply Cauchy–Schwarz inequality. Similarly, term $\left\|\nabla_{\mathbf{x}}\mathcal{L}(\mathbf{x}^{t+1},\mathbf{y}^{t+1},\mathbf{z}^t)\right\|^2$ can be bound as:

$$
\begin{aligned}
\left\|\nabla_{\mathbf{x}}\mathcal{L}(\mathbf{x}^{t+1},\mathbf{y}^{t+1},\mathbf{z}^t)\right\|^2 &\overset{(a)}{\leq} 2\left\|\nabla_{\mathbf{x}}\mathcal{L}(\mathbf{x}^{t+1},\mathbf{y}^{t+1},\mathbf{z}^t) - \nabla_{\mathbf{x}}\mathcal{L}(\mathbf{x}^{t+1},\mathbf{y}^{t+1},\mathbf{z}^{t+1})\right\|^2 \\
&\quad + 2\left\|\nabla_{\mathbf{x}}\mathcal{L}(\mathbf{x}^{t+1},\mathbf{y}^{t+1},\mathbf{z}^{t+1})\right\|^2 \\
&\overset{A4}{\leq} 2C_{\mathbf{x}}^2\left\|\mathbf{z}^{t+1}-\mathbf{z}^t\right\|^2 + 2\left\|\nabla_{\mathbf{x}}\mathcal{L}(\mathbf{x}^{t+1},\mathbf{y}^{t+1},\mathbf{z}^{t+1})\right\|^2,
\end{aligned}
\tag{33}
$$

where in $(a)$ we add and subtract $\nabla_{\mathbf{x}}\mathcal{L}(\mathbf{x}^{t+1},\mathbf{y}^{t+1},\mathbf{z}^{t+1})$ and apply Cauchy–Schwarz inequality.

Then we sum the above results as $(24)\times 2 + (32)\times\frac{\eta}{5} + (33)\times\frac{\eta}{5}$ and obtain the following:

$$
\begin{aligned}
&2\,\mathbb{E}^t\,\mathcal{L}(\mathbf{x}^{t+1},\mathbf{y}^{t+1},\mathbf{z}^{t+1}) - 2\mathcal{L}(\mathbf{x}^t,\mathbf{y}^t,\mathbf{z}^t) + \frac{\eta}{5}\,\mathbb{E}^{t,Q-1}\left\|\nabla_{\mathbf{x}}\mathcal{L}(\mathbf{x}^{t+1},\mathbf{y}^t,\mathbf{z}^t)\right\|^2 \\
&+ \frac{\eta}{5}\left\|\nabla_{\mathbf{x}}\mathcal{L}(\mathbf{x}^{t+1},\mathbf{y}^{t+1},\mathbf{z}^t)\right\|^2 \leq -\eta\sum_{q=0}^{Q-1}\mathbb{E}^t\left\|\nabla_{\mathbf{x}}\mathcal{L}(\mathbf{x}^{t,q},\mathbf{y}^t,\mathbf{z}^t)\right\|^2 - \mu\left\|\mathbf{z}^{t+1}-\mathbf{z}^t\right\|^2 \\
&+ QL_{\mathbf{x}}\eta^2\sigma^2 + \frac{2C_{\mathbf{x}}^2\eta}{5}\left\|\mathbf{z}^{t+1}-\mathbf{z}^t\right\|^2 + \frac{2\eta}{5}\left\|\nabla_{\mathbf{x}}\mathcal{L}(\mathbf{x}^{t+1},\mathbf{y}^{t+1},\mathbf{z}^{t+1})\right\|^2 \\
&+ \frac{2\eta+2L_{\mathbf{x}}^2\eta^3}{5}\left\|\nabla_{\mathbf{x}}\mathcal{L}(\mathbf{x}^{t,Q-1},\mathbf{y}^t,\mathbf{z}^t)\right\|^2 + \frac{2L_{\mathbf{x}}^2\eta^3\sigma^2}{5}.
\end{aligned}
\tag{34}
$$

Rearrange the terms, notice that we choose $\eta\leq\frac{1}{L_{\mathbf{x}}}$, so that $2\eta+2L_{\mathbf{x}}^2\eta^3\leq 4\eta$, we have:

$$
\begin{aligned}
&\frac{\eta}{5}\sum_{q=0}^{Q}\mathbb{E}^t\left\|\nabla_{\mathbf{x}}\mathcal{L}(\mathbf{x}^{t,q},\mathbf{y}^t,\mathbf{z}^t)\right\|^2 + \frac{\eta}{5}\,\mathbb{E}^t\left\|\nabla_{\mathbf{x}}\mathcal{L}(\mathbf{x}^{t+1},\mathbf{y}^{t+1},\mathbf{z}^{t+1})\right\|^2 + \left(\mu - \frac{2C_{\mathbf{x}}^2\eta}{5}\right)\left\|\mathbf{z}^{t+1}-\mathbf{z}^t\right\|^2 \\
&\leq 2\left(\mathcal{L}(\mathbf{x}^t,\mathbf{y}^t,\mathbf{z}^t) - \mathbb{E}^t\,\mathcal{L}(\mathbf{x}^{t+1},\mathbf{y}^{t+1},\mathbf{z}^{t+1})\right) + \left(\frac{2L_{\mathbf{x}}^2\eta^3}{5} + QL_{\mathbf{x}}\eta^2\right)\sigma^2.
\end{aligned}
\tag{35}
$$

Sum the above equation from $t=0$ to $T-1$, choose $\mu - \frac{2C_{\mathbf{x}}^2\eta}{5}\geq\frac{\mu}{5}$ $\left(\eta\leq\frac{8\mu}{5C_{\mathbf{x}}^2}\right)$, and devide both side by $\frac{\eta QT}{5}$, then Theorem 2 is proved.

## B.2 Proof for Lemma 1

In this section, we verify the assumptions A4-A9 for the original problem (6) under assumptions A1-A3.

Recall that we have the following correspondence:

$$\mathbf{x} := [\Theta_1; \ldots; \Theta_M; w_1; \ldots; w_M], \quad \mathbf{y} := [\text{vec}(\Pi_1); \ldots, \text{vec}(\Pi_M)], \quad \mathbf{z} := [\Theta_0; w_0],$$

$$\mathcal{L}(\mathbf{x}, \mathbf{y}, \mathbf{z}) := \sum_{m=1}^{M} p_m \left( f_m(\Theta_m, w_m) + \mu_1 \cdot r_{m,1}(\Theta_m, \Theta_0) + \mu_2 \cdot r_{m,2}(w_m, \Pi_m, w_0) \right),$$

$$r_{m,1}(\Theta_m, \Theta_0) = \frac{1}{2} \left\| \Theta_m - P_m \Theta_0 \right\|^2,$$

$$r_{m,2}(w_m, \Pi_m, w_0) = \frac{1}{2} \left\| w_m - \Pi_m w_0 \right\|^2, \quad \text{s.t.} \quad \Pi_m \in \mathcal{S}(d_{w,m}, d_{w,0}), \ \sum_{m=1}^{M} \mathbf{1}_{d_{w,m}}^T \Pi_m \geq \mathbf{1}_{d_{w,0}}^T.$$

**1)** For A4, we have

$$\nabla_{\mathbf{x}} \mathcal{L}(\mathbf{x}, \mathbf{y}, \mathbf{z}) = \left[ \begin{array}{c} p_m \nabla_{\Theta_m} f_m(\Theta_m, w_m) + p_m \mu_1 (\Theta_m - P_m \Theta_0) \\ p_m \nabla_{w_m} f_m(\Theta_m, w_m) + p_m \mu_2 (w_m - \Pi_m w_0) \end{array} \right]_{m=1}^{M}.$$

Therefore we have the following bound:

$$\left\| \nabla_{\mathbf{x}} \mathcal{L}(\mathbf{x}, \mathbf{y}, \mathbf{z}) - \nabla_{\mathbf{x}} \mathcal{L}(\mathbf{x}', \mathbf{y}, \mathbf{z}) \right\|$$

$$= \sum_{m=1}^{M} p_m \left\| \nabla_{\Theta_m} f_m(\Theta_m, w_m) + \mu_1 \Theta_m - \nabla_{\Theta_m} f_m(\Theta_m', w_m') - \mu_1 \Theta_m' \right\|$$

$$+ \sum_{m=1}^{M} p_m \left\| \nabla_{w_m} f_m(\Theta_m, w_m) + \mu_2 w_m - \nabla_{w_m} f_m(\Theta_m', w_m') - \mu_2 w_m' \right\|$$

$$\leq \sum_{m=1}^{M} p_m \left( \left\| \nabla_{\Theta_m} f_m(\Theta_m, w_m) - \nabla_{\Theta_m} f_m(\Theta_m', w_m') \right\| + \mu_1 \left\| \Theta_m - \Theta_m' \right\| \right)$$

$$+ \sum_{m=1}^{M} p_m \left( \left\| \nabla_{w_m} f_m(\Theta_m, w_m) - \nabla_{w_m} f_m(\Theta_m', w_m') \right\| + \mu_2 \left\| w_m - w_m' \right\| \right)$$

$$\overset{A1}{\leq} \sum_{m=1}^{M} p_m \left( (L_m + \mu_1) \cdot \left\| \Theta_m - \Theta_m' \right\| + (L_m + \mu_2) \cdot \left\| w_m - w_m' \right\| \right)$$

$$\leq \max_m \{ p_m L_m + \max\{\mu_1, \mu_2\} \} \sum_{m=1}^{M} \left( \left\| \Theta_m - \Theta_m' \right\| + \left\| w_m - w_m' \right\| \right)$$

$$= L_{\mathbf{x}} \left\| \mathbf{x} - \mathbf{x}' \right\|.$$

where we obtain $L_{\mathbf{x}} = \max_m \{ p_m L_m + \max\{\mu_1, \mu_2\} \}$. Also, we have

$$\left\| \nabla_{\mathbf{x}} \mathcal{L}(\mathbf{x}, \mathbf{y}, \mathbf{z}) - \nabla_{\mathbf{x}} \mathcal{L}(\mathbf{x}, \mathbf{y}, \mathbf{z}') \right\| = \sum_{m=1}^{M} p_m \left( \mu_1 \left\| P_m(\Theta_0 - \Theta_0') \right\| + \mu_2 \left\| \Pi_m(w_0 - w_0') \right\| \right)$$

$$\leq \sum_{m=1}^{M} p_m \left( \mu_1 \left\| P_m \right\| \left\| \Theta_0 - \Theta_0' \right\| + \mu_2 \left\| \Pi_m \right\| \left\| w_0 - w_0' \right\| \right)$$

$$\overset{(a)}{=} \sum_{m=1}^{M} p_m \left( \mu_1 \left\| \Theta_0 - \Theta_0' \right\| + \mu_2 \left\| w_0 - w_0' \right\| \right)$$

$$\overset{(b)}{\leq} \max\{\mu_1, \mu_2\} \left( \left\| \Theta_0 - \Theta_0' \right\| + \left\| w_0 - w_0' \right\| \right)$$

$$= C_{\mathbf{x}} \left\| \mathbf{z} - \mathbf{z}' \right\|.$$

where in $(a)$ we use the fact that $P_m \in \mathcal{S}(d_m, d_0), \Pi_m \in \mathcal{S}(d_{w,m}, d_{w,0})$ are selection matrices so that $\| P_m \| = 1, \| \Pi_m \| = 1$; $(b)$ uses the fact that $\sum_{m=1}^{M} p_m = 1$. Therefore A4 is verified.

**2)** Next, we verify A5. We proceed by directly computing the second derivitive of $\mathbf{z}$:

$$
\nabla_{\mathbf{z}}^2 \mathcal{L}(\mathbf{x}, \mathbf{y}, \mathbf{z})
$$

$$
= \sum_{m=1}^{M} p_m \begin{bmatrix} \nabla_{\Theta_0}^2 \mu_1 \cdot r_{m,1} + \mu_2 \cdot r_{m,2} & \nabla_{\Theta_0} \nabla_{w_0} \mu_1 \cdot r_{m,1} + \mu_2 \cdot r_{m,2} \\ \nabla_{\Theta_0} \nabla_{w_0} \mu_1 \cdot r_{m,1} + \mu_2 \cdot r_{m,2} & \nabla_{w_0}^2 \mu_1 \cdot r_{m,1} + \mu_2 \cdot r_{m,2} \end{bmatrix}
$$

$$
= \begin{bmatrix} \mu_1 \cdot \sum_{m=1}^{M} p_m P_m^T P_m & 0 \\ 0 & \mu_2 \sum_{m=1}^{M} p_m \Pi_m^T \Pi_m \end{bmatrix}.
$$

Then we analyze the range of the eigenvalues of this matrix. First we know that $\Pi_m, P_m$'s are selection matricies, therefore $P_m^T P_m, \Pi_m^T \Pi_m$ are diagonal matricies, indicating that $\nabla_{\mathbf{z}}^2 \mathcal{L}(\mathbf{x}, \mathbf{y}, \mathbf{z})$ is also a diagonal matrix.

For the first block $\mu_1 \cdot \sum_{m=1}^{M} p_m P_m^T P_m$, we have that $P_m$'s are the feature selection matrix, i.e.,

$$
x_m = P_m x_0, \quad x_m \in \mathcal{X}_m = \prod_{i \in \mathcal{I}_m} \mathcal{D}_i, \quad x_0 \in \mathcal{X}_0 = \prod_{i=1}^{d_0} \mathcal{D}_i.
$$

It is clear that if client $m$ has the $i^{\text{th}}$ feature, then the $i^{\text{th}}$ diagonal entry of $P_m^T P_m$ is $P_m^T P_m(i,i) = 1$, and $P_m^T P_m(i,i) = 0$ otherwise. That is, the following holds:

$$
P_m^T P_m(i,i) = \begin{cases} 1, i \in \mathcal{I}_m, \\ 0, i \notin \mathcal{I}_m. \end{cases}
$$

Further, we have that the full feature space $\mathcal{X}_0$ is the union of the clients' feature spaces, i.e., $\bigcup_{m \in [M]} \mathcal{I}_m = [d_0]$. Therefore, we have

$$
\mathbb{1}_{d_0}^T \sum_{m=1}^{M} P_m^T P_m \geq \mathbb{1}_{d_0}^T, \text{ and } \min_{i \in [d_0]} \sum_{m=1}^{M} p_m P_m^T P_m(i,i) \geq \min_{m} \{p_m\}.
$$

Similarly, the constraint on $\Pi_m$'s that $\sum_{m=1}^{M} \mathbf{1}_{d_{w,m}}^T \Pi_m \geq \mathbf{1}_{d_{w,0}}^T$ indicates that the following holds:

$$
\min_{i \in [d_{m,0}]} \sum_{m=1}^{M} p_m \Pi_m^T \Pi_m(i,i) \geq \min_{m} \{p_m\}.
$$

Therefore, the Hessien matrix $\nabla_{\mathbf{z}}^2 \mathcal{L}(\mathbf{x}, \mathbf{y}, \mathbf{z})$ is positive definite, with smallest eigenvalue $\mu \geq \min_{m} \{p_m\} \cdot \min\{\mu_1, \mu_2\}$. Thus A5 is verified.

**3)** A6 holds true as in Algorithm 1, we uniformly samples $n \in \mathcal{N}_m$ for all $m \in [M]$, therefore

$$
\mathbb{E}_n \nabla_{\Theta_m} \ell(F_m(w_m; H_m(\Theta_m; x_{m,n})), y_n) = \nabla_{\Theta_m} f_m(\Theta_m, w_m),
$$

$$
\mathbb{E}_n \nabla_{w_m} \ell(F_m(w_m; H_m(\Theta_m; x_{m,n})), y_n) = \nabla_{w_m} f_m(\Theta_m, w_m),
$$

Further, from A3, we can obtain A7 with $\sigma^2 = \sigma_\Theta^2 + \sigma_w^2$.

**4)** To verify A8, we apply A2 that:

$$
\mathcal{L}(\mathbf{x}, \mathbf{y}, \mathbf{z}) = \sum_{m=1}^{M} p_m \left( f_m(\Theta_m, w_m) + \mu_1 \cdot r_{m,1}(\Theta_m, \Theta_0) + \mu_2 \cdot r_{m,2}(w_m, \Pi_m, w_0) \right)
$$

$$
\overset{(a)}{\geq} \sum_{m=1}^{M} p_m f_m(\Theta_m, w_m) \overset{A2}{\geq} \sum_{m=1}^{M} p_m \underline{f_m} := \underline{\mathcal{L}},
$$

| Client index $m$ | Assigned features $\mathcal{I}_m$ | Assigned class # |
|---|---|---|
| 1 | 1,2,3 | 25 |
| 2 | 1,2,3 | 25 |
| 3 | 1,3,4 | 25 |
| 4 | 1,3,4 | 25 |
| 5 | 1,3 | 25 |
| 6 | 1,3 | 25 |

Table 3: The data assignment pattern for MultiView40 dataset. Note that 6.88% of data has never been used.

| Client index $m$ | Assigned features $\mathcal{I}_m$ | Assigned classes |
|---|---|---|
| 1 | 1,2,3 | 1–5 |
| 2 | 1,2,3 | 6–10 |
| 3 | 1,3,4 | 1–5 |
| 4 | 1,3,4 | 6–10 |
| 5 | 1,3 | 1–5 |
| 6 | 1,3 | 6–10 |

Table 4: The data assignment pattern for Cifar-10 and EuroSAT dataset.

where $(a)$ uses the fact that $r_{m,1}(\Theta_m, \Theta_0) = \frac{1}{2}\|\Theta_m - P_m\Theta_0\|^2 \geq 0$ and $r_{m,2}(w_m, \Pi_m, w_0) = \frac{1}{2}\|w_m - \Pi_m w_0\|^2 \geq 0$.

**5)** A9 directly comes from the constraint on $\Pi_m$'s that

$$\Pi_m \in \mathcal{S}(d_{w,m}, d_{w,0}), \ \sum_{m=1}^{M} \mathbf{1}_{d_{w,m}}^T \Pi_m \geq \mathbf{1}_{d_{w,0}}^T,$$

which is a compact set.

At this point, we have verified A4-A9 for problem (6) with Algorithm 1, and the corresponding constants are summarized below:

$$L_{\mathbf{x}} = \max_m\{p_m L_m + \max\{\mu_1, \mu_2\}\}, \ C_{\mathbf{x}} = \max\{\mu_1, \mu_2\},$$

$$\mu \geq \min_m\{p_m\} \cdot \min\{\mu_1, \mu_2\}, \ \sigma^2 = \sigma_\Theta^2 + \sigma_w^2, \ \underline{\mathcal{L}} = \sum_{m=1}^{M} p_m \underline{f_m}.$$

This completes the proof for Lemma 1.

## C   Additional Numerical Experiments

In this section, we include additional sets of numerical experiments. In the first set of additional experiments, we reduce the feature heterogeneity of the data on the clients by allowing clients to have common features so that HFL algorithms such as FedProx and FedAvg apply. Further, we include an additional multi-modal dataset with both image and text features.

### C.1   Comparison with HFL

In this section, we conduct numerical experiments to compare FedProx (Li et al., 2018) with HyFEM. In the experiments, we split the features into $d_0 = 4$ blocks for the datasets and assign the first and the third blocks as the common blocks for all $M = 6$ clients. Then we can apply FedProx to train a model with the overlapped features and compare with the models trained with HyFEM with more features. The detailed data assignment patterns for different datasets are described in Table 3 - 4. Note that in the MultiView40 dataset, there are 6.88% of the data has never been used by HyFEM and 50% of the data has never been

used by FedProx. For Cifar-10 and EuroSAT datasets, all data has been used by at least one client with HyFEM while 50% of the data are dropped by FedProx.

For Cifar-10 and EuroSAT datasets, we split each image into (top left, top right, bottom left, bottom right) total $d_0 = 4$ feature blocks. For the MultiView40 dataset, we choose four views of different angles of the objects as the full feature space. The total communication round is $T = 64$ and local update # $Q = 32$ are fixed for all experiments. We conducted a line search on the learning rate $\eta$ and $\mu_2$ for the algorithms to obtain the best performance.

The results for the datasets are shown in Figure 9 and Figure 10. From the results, we can see that the models trained with HyFEM can obtain better performance than FedProx. This is because HyFEM can use more data than FedProx by using heterogeneous models.

## C.2 Data Partitioning Pattern

In this subsection, we provide the data partitioning patterns for each setting. Note that in Figure 11(b), and Figure 12(a), the black boxes with 0 inside them indicate that the corresponding feature block of the samples in this class has not been used for training by any of the clients.

## C.3 Experiments on HeriGraph Dataset

HeriGraph (Bai et al., 2022) dataset is a multi-modal dataset for heritage site classification. This dataset consists of total $\mathcal{N}_0 = 41,621$ samples from 9 classes. Each sample has at most $d_0 = 4$ preprocessed feature blocks, including one text feature block and three different image feature blocks. Note that not all samples have all features. For example, only $25,325$ samples have text features, and the rest do not. We will include the result for this dataset in the revised manuscript.

In the experiments, we use the MLP model with one hidden layer as the classifier $f_m(w_m; \cdot)$. We use MLPs of different sizes as feature extractors for each feature block. We set client number $M = 6$, client feature block number $d_m = 2$, and each client has 6 out of 9 classes.

The result is shown in Figure 13. The server model trained with HyFEM has comparable performance as the centralized trained model. The average performance of the clients' models has worse performance than the full model due to the lack of full features and classes, but the accuracy is 20% higher than the models obtained with stand-alone training.

## C.4 Ablation Study on Reqularizer Weight

In this subsection, we provide extra numerical results on different choices of the regularizer weight $\mu_2$. In the experiment, we trained the model on the MultiView40 dataset under setting MultiView40:2. In the experiment, we choose $\mu_2 = \{0.1, 0.3, 0.5\}$ to control the strength of the impact of model matching and the divergence between the local and global model. The results are shown in Figure **??** From the result, we can see that with larger $\mu_2$, i.e., when regularizing the client and the matched server model to be more similar, both the server-side and the client-side models achieve better performance.

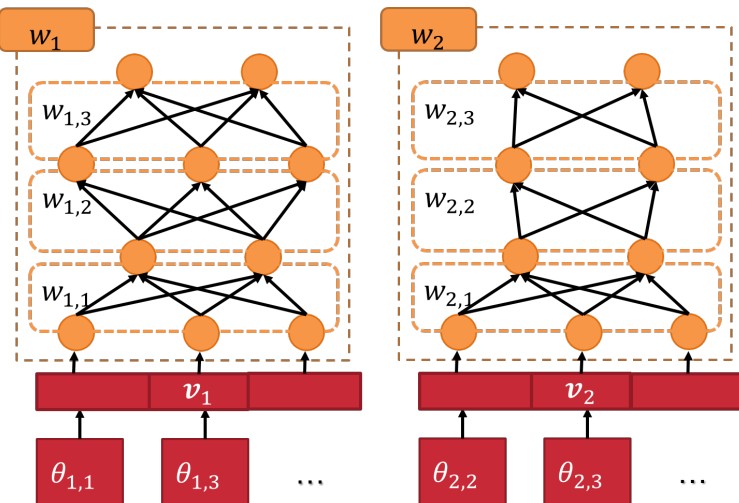

Figure 6: Illustration of the layer structure of the inference blocks on the clients, for the $L = 3$ case.

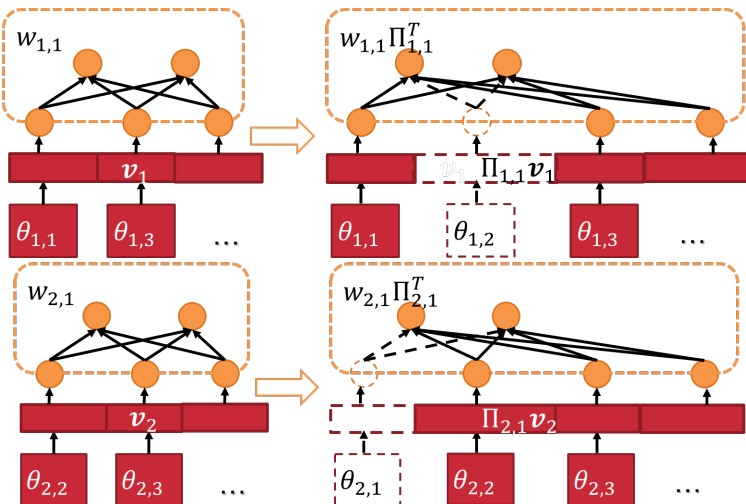

Figure 7: Aligning the input of the first layer by rearranging and padding the corresponding coordinates of the input $\mathbf{v}_m$ and the first layer $w_{m,1}$.

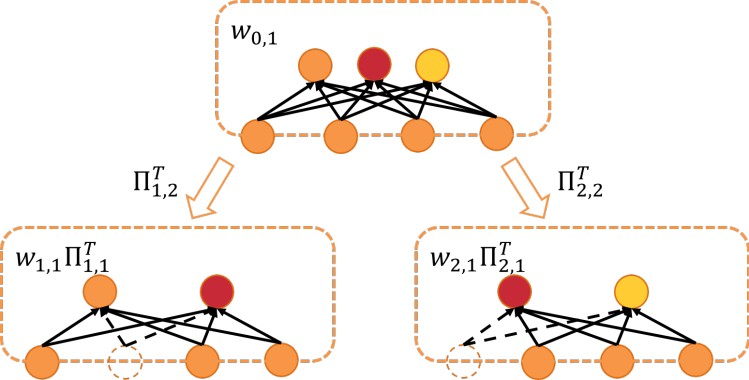

Figure 8: Aligning the output of the first layer by rearranging and padding the corresponding coordinates of the first layer $w_{m,1}$.

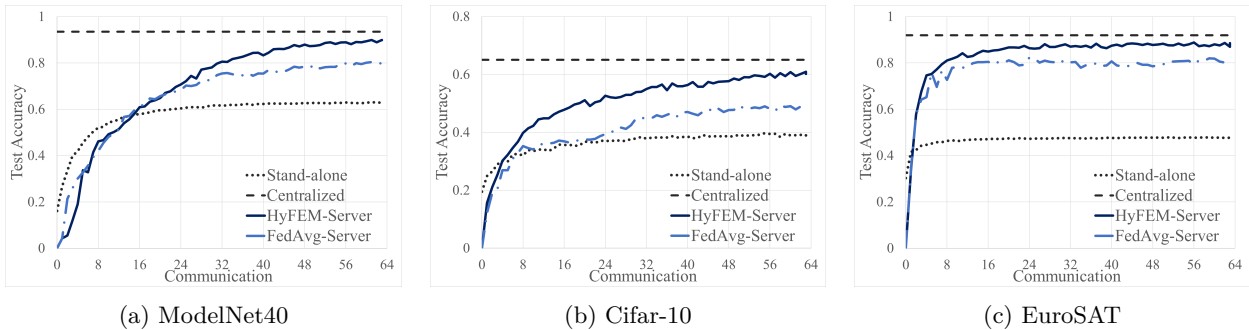

(a) ModelNet40      (b) Cifar-10      (c) EuroSAT

Figure 9: Test accuracy of server models trained with HyFEM compared with FedProx on a) ModelNet40, b) Cifar-10, and c) EuroSAT.

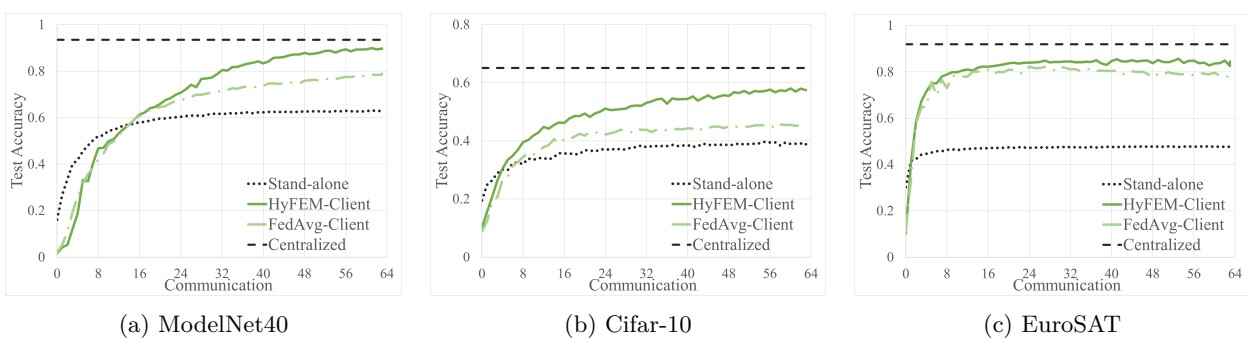

(a) ModelNet40      (b) Cifar-10      (c) EuroSAT

Figure 10: Averaged test accuracy of client models trained with HyFEM compared with FedProx on a) ModelNet40, b) Cifar-10, and c) EuroSAT.

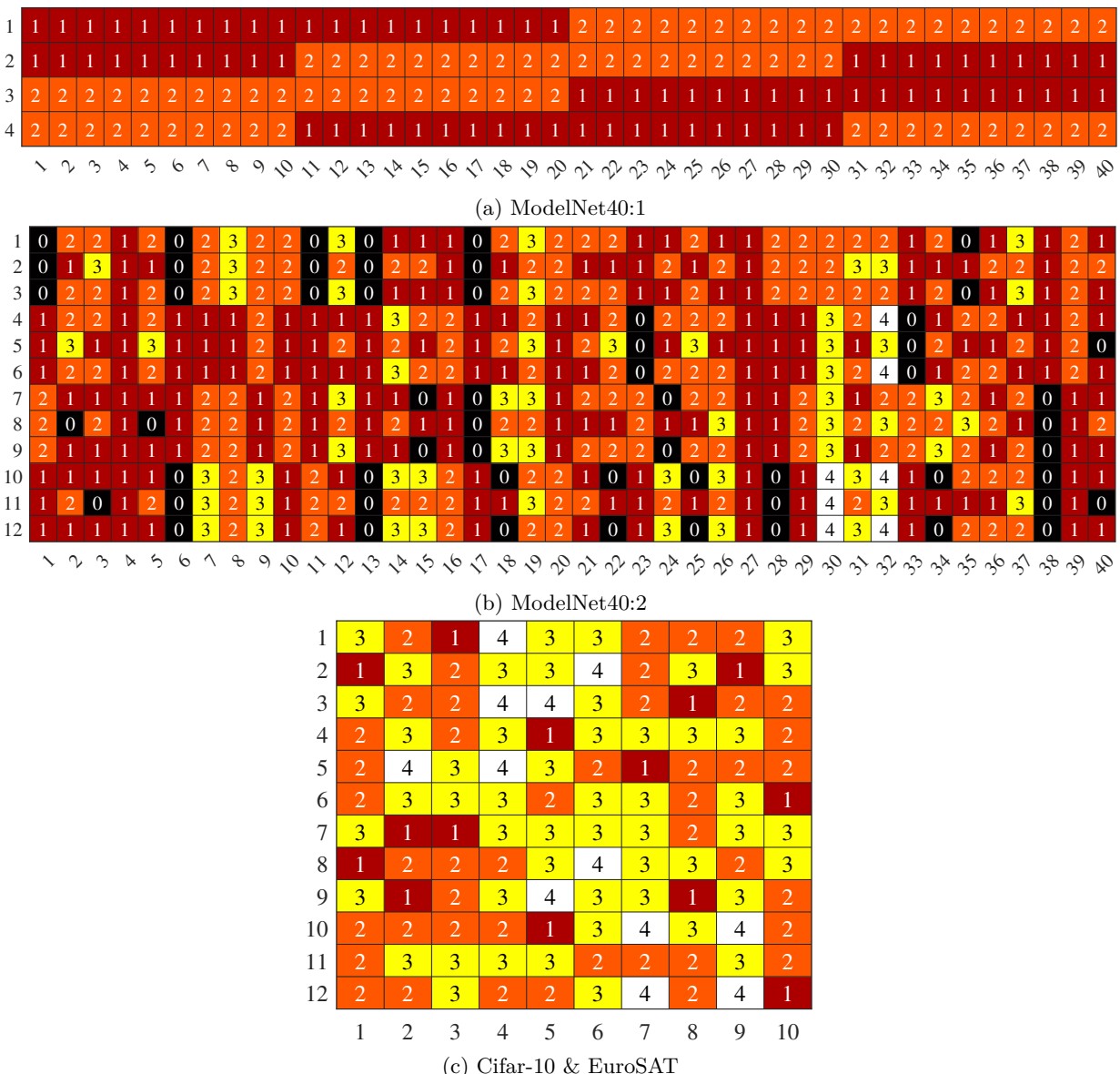

Figure 11: The illustration of how many clients (the numbers in boxes) possess the training data of each feature block in each class for the settings in Section 4, with a) ModelNet40:1 with $d_0 = 4$ features, $M = 4$ clients, and 40 classes; b) ModelNet40:2 with $d_0 = 12$ features, $M = 8$ clients, and 40 classes; c) Cifar-10 & EuroSAT with $d_0 = 12$ features, $M = 9$ clients, and 10 classes. The $x$-axis of each plot is the class axis and $y$-axis is the feature axis.

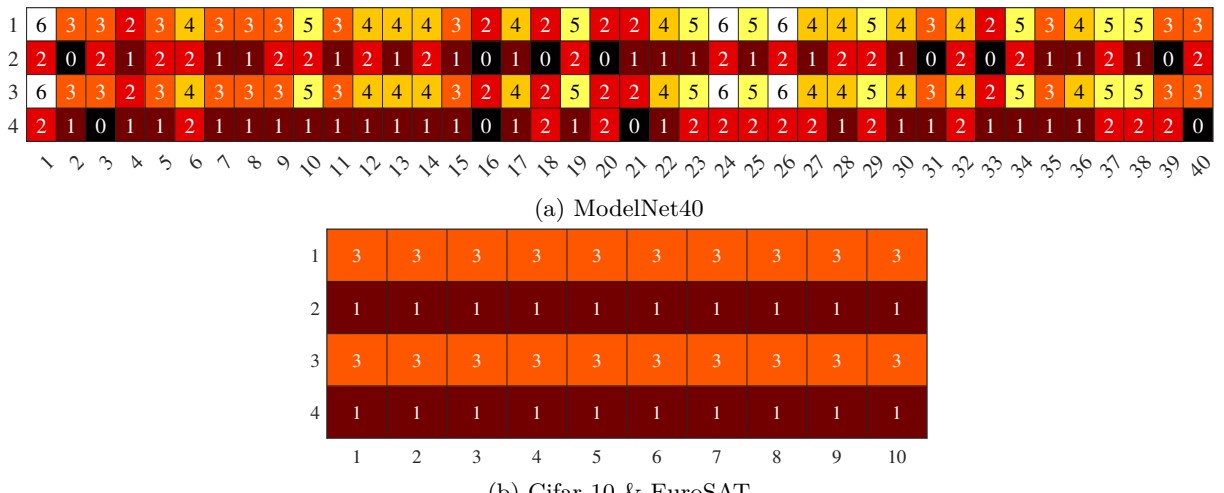

(a) ModelNet40

(b) Cifar-10 & EuroSAT

Figure 12: The illustration of how many clients (the numbers in boxes) possess the training data of each feature block in each class for the settings in Appendix C.1, with a) ModelNet40 with $d_0 = 4$ features, $M = 6$ clients, and 40 classes; c) Cifar-10 & EuroSAT with $d_0 = 4$ features, $M = 6$ clients, and 10 classes. The $x$-axis of each plot is the class axis and $y$-axis is the feature axis.

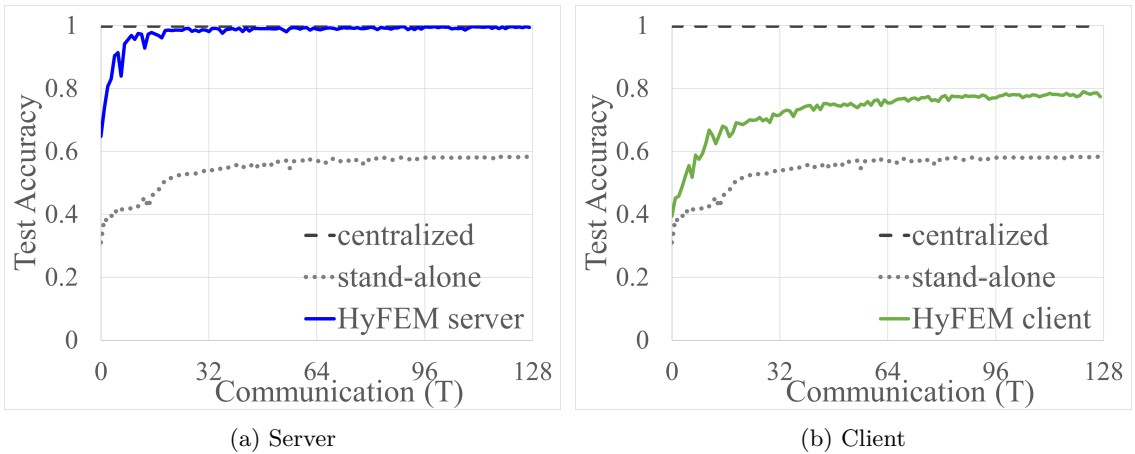

(a) Server

(b) Client

Figure 13: Test accuracy of a) server model, b) client models trained with HyFEM compared with Centralized training and stand-alone training for HeriGraph dataset.

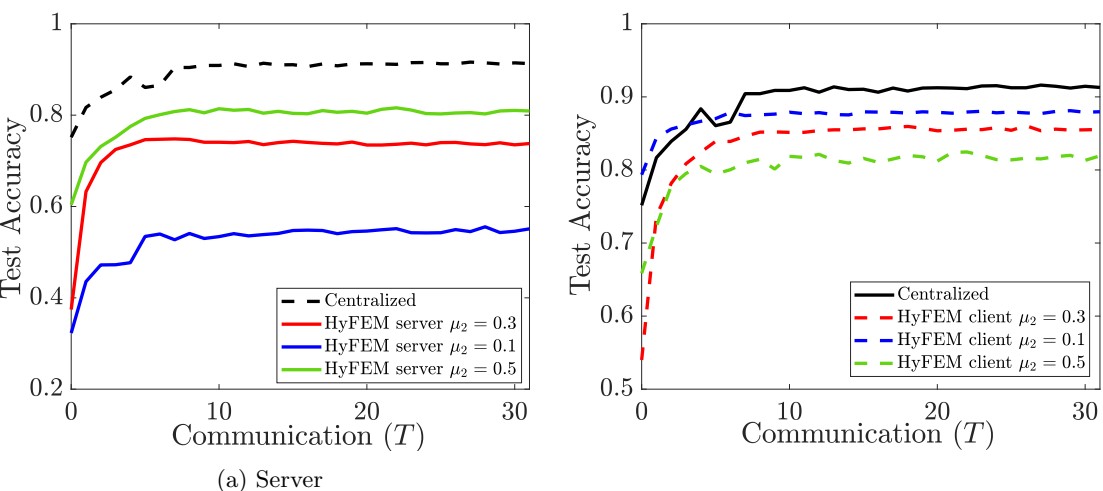

(a) Server

Figure 14: Test accuracy of a) server model, b) client models trained with HyFEM compared with Centralized training on ModelNet40:2 under different $\mu_2$.

