# OpenReview forum: "Hybrid Federated Learning for Feature & Sample Heterogeneity: Algorithms and Implementation"
_TMLR — Accepted by TMLR_

### Review · Reviewer_w1tn · 2024-02-06

**Summary Of Contributions:**

The paper proposes an approach to a hybrid federated learning setting that combines horizontal and vertical federated learning: clients observe a subset of data points and features. Clients train feature extractors for each possible subset of features in the federation and a classifier on the concatenation of features produced by all locally applicable feature extractors. For collaboration, the server extracts server-side feature extractors and a classifier, local clients regularize their local feature extractors and classifiers to be similar to the server one.

The paper provides a theoretical convergence analysis and an empirical evaluation and it shows how the proposed approach reduces to established algorithms in their respective settings (e.g., only horizontal or vertical FL).

**Audience:**

Yes

**Broader Impact Concerns:**

I do not have any concerns on the ethical implications of this work and therefore think a broader impact statement is unnecessary.

**Claims And Evidence:**

Yes

**Requested Changes:**

**The following changes are small, but critical in my view:**
- A more thorough discussion of related work seems appropriate. In particular, a discussion of VFL methods and their applicability to this setting, and split learning seems appropriate.
- The definition of feature heterogeneity as clients observing various subsets is a bit ambiguous, since the term is also sometimes used for feature shift, i.e., clients observing the same features but the distribution of features being different. It would be great to briefly discuss this difference. One could even argue that the proposed approach is compatible with existing work on feature shift (e.g., FedBN [1]), which further strengthens its significance.
- The empirical evaluation shows promising results, but lacks a significance and ablation study. I suggest adding experiments on the impact of the communication period, and the regularization parameters $\mu_1, \mu_2$. As an ablation study, I suggest investigating only training the classifier collaboratively, e.g., by setting $\mu_1=0$.
- In Fig. 5, please add the variance of client accuracies, or the individual client accuracies.


**The following changes would strengthen this work:**
- In its current formulation, the approach uses feature extractors for all feature subsets observed by clients. One could imagine that it is advantageous to go a step further: one could construct a feature extractor for every possible subset of features, i.e., the power set without the empty set. While this might be not applicable to image data, it certainly could work for tabular data. A comparison of the current approach to this rather extreme version would be very interesting.
- The amount of clients used is rather small ($M=6$). Since the paper motivates their approach with a medical scenario, I think this is a realistic choice. However, it would be great to see how well the approach scales to larger numbers of clients.
- In semi-supervised federated learning it is quite common to use different local models at each client. While this setting is different from the hybrid FL setting considered in this paper, I suggest adding a very brief discussion about the potential applicability of semi-supervised FL methods to this setting under the assumption that public (unlabeled) data exists, for example using distributed distillation [2] or federated co-training [3].
- Out of curiousity: would this approach be applicable to kernel methods, as well? Here, one would typically exchange support vectors and their corresponding coefficients (cf. [4]), so "model matching" would probably translate to "support vector matching". It would be interesting to discuss this, as well.


[1] Li, Xiaoxiao, et al. "FedBN: Federated Learning on Non-IID Features via Local Batch Normalization." International Conference on Learning Representations. 2021.
[2] Bistritz, Ilai, Ariana Mann, and Nicholas Bambos. "Distributed distillation for on-device learning." Advances in Neural Information Processing Systems 33 (2020): 22593-22604.
[3] Abourayya, Amr, et al. "Protecting Sensitive Data through Federated Co-Training." arXiv preprint arXiv:2310.05696 (2023).
[4] Kamp, Michael, et al. "Communication-efficient distributed online learning with kernels." Machine Learning and Knowledge Discovery in Databases. Springer International Publishing, 2016.

**Strengths And Weaknesses:**

**Strengths:**
- novel approach to a relevant problem
- sound method that is well-analyzed theoretically
- promising empirical results

**Weaknesses:**
- empirical evaluation is limited
- approach is somewhat straight-forward

---

> ### Author Response · Authors · 2024-03-08
> **Reply to reviewer w1tn (Part I)**
>
> 1. **Related work:**
>     1. Hybrid Federated Learning (Hybrid FL) can refer to different settings in the existing literature. In [R1], it refers to the setting where the training is a hybridization of centralized learning and federated learning, i.e., the training samples are held partially by the server and partially distributed to the clients. In [R2], Hybrid FL for sensor network refers to a hierarchical federated learning system, where each client consists of multiple sensors with vertically distributed data, and the clients are connected to a server to perform horizontal FL for distributed sensing. Although named as hybrid FL, the above-mentioned settings are different from the settings on our paper, i.e., each client holds partial samples and partially overlapping features. Therefore, these algorithms cannot be applied to our setting.
>     2. Federated contrastive learning (FedCL) is another set of algorithms related to Hybrid FL. In this setting, the clients hold non-overlapping features and partially overlapping samples and aim to learn separate models for local inference (e.g., [R3, R4]). The algorithms perform vertical FL on the overlapping samples to train a global guidance model and perform local contrastive learning (self-supervised learning) with the non-overlapped local data to train local models. Compared with our setting, the FedCL algorithm requires overlapping samples and transmitting intermediate features as VFL and fails to make use of overlapping feature spaces and non-overlapping samples during VFL training.
>     3. Feature shift (a.k.a. domain shift) in FL considers the situation where the clients' features have a different distribution, e.g., the shift from oil painting image to sketch or comic style, while the shifted feature type and dimension are the same as the original feature. Feature shift can happen in both training procedures across different clients [R5] and the deployment phase, where training data and inference data come from different distributions [R6].  The feature heterogeneity in feature shift is different from the ones in hybrid FL in this paper. In hybrid FL, different feature blocks can have heterogeneous distributions and *types*, e.g., table, text, image, or time series. Therefore, hybrid FL covers more complex data/feature distributions.
> 2. **Ablation study:** We provide an additional ablation study on the weights of the classifier's regularizer $\mu_2$; we will add the result in the revised appendix C.4.
> 3. **Variance of client accuracy in Fig. 5:** Thanks for the suggestion, we will add the variance of the clients' accuracy in the revised manuscript.
> 4. **More feature extractor:** It is natural to consider the extreme case of training feature extractors that cover the power set of the feature blocks. However, such an approach is impractical for two reasons: 1) the lack of necessity and 2) large resource consumption. Training on the power set of the feature blocks can be unnecessary. In the case where the feature blocks have different modalities (e.g., text and image or MRI and CT results), it is impossible to design such a feature extractor that takes the combination of different features as input. Second, the power set grows exponentially as the feature block number. For example, in our experiment on the ModelNet40 dataset with 12 feature blocks, training on the power set requires training on $2^{12}-1=4095$ feature extractors, which takes around $400\times$ more computational resources than the current strategy that trains separate feature extractors for each block.

---

> ### Author Response · Authors · 2024-03-08
> **Reply to reviewer w1tn (Part II)**
>
> 5. **Scaling with client number $M$:** In our experiment, we use client number $M=4, 6, 8, 9$ for different settings and datasets. Although it is interesting to see how the algorithm scales with larger client number, but we believe that the current experiment setting is sufficient for the validation of the proposed algorithm in a reasonable setting, and we will differ the scaling to larger client number as a future work.
> 6. **Adapting to kernel method:** This is an interesting direction. As the reviewer suggested, one can probably match the kernels by their similarity. However, the extension to the kernel method is a non-trivial extension. Specifically, the kernels of the local clients serve as feature extractors in the hybrid FL setting, and the clients require additional post-processing layers (e.g., classifier or regression layer) for different tasks. So, the matching procedure of the client kernels plays a similar role in feature alignment in hybrid FL, which does not exist in the current HyFEM algorithm. Nevertheless, we agree with the reviewer that the extension to federated kernel methods with kernel matching could be an interesting future direction.
>
> [R1] Elbir, A. M., Coleri, S., Papazafeiropoulos, A. K., Kourtessis, P., & Chatzinotas, S. (2022). A hybrid architecture for federated and centralized learning. IEEE Transactions on Cognitive Communications and Networking, 8(3), 1529-1542.
>
> [R2] Su, L., & Lau, V. K. (2021). Hierarchical federated learning for hybrid data partitioning across multitype sensors. IEEE Internet of Things Journal, 8(13), 10922-10939.
>
> [R3] He, Y., Kang, Y., Zhao, X., Luo, J., Fan, L., Han, Y., & Yang, Q. (2022). A hybrid self-supervised learning framework for vertical federated learning. arXiv preprint arXiv:2208.08934.
>
> [R4] Kang, Y., Liu, Y., & Liang, X. (2022). FedCVT: Semi-supervised vertical federated learning with cross-view training. ACM Transactions on Intelligent Systems and Technology (TIST), 13(4), 1-16.
>
> [R5] Huang, W., Ye, M., & Du, B. (2022). Learn from others and be yourself in heterogeneous federated learning. In Proceedings of the IEEE/CVF Conference on Computer Vision and Pattern Recognition (pp. 10143-10153).
>
> [R6] Zhang, L., Lei, X., Shi, Y., Huang, H., & Chen, C. (2021). Federated learning with domain generalization. arXiv preprint arXiv:2111.10487.

---

### Review · Reviewer_yZ9H · 2024-02-12

**Summary Of Contributions:**

This study explores a unique aspect of federated learning known as hybrid federated learning (FL), characterized by each client possessing a different subset of features and data samples. Unlike conventional FL scenarios, the hybrid FL framework introduces d_0 feature blocks, with each client holding a portion of these blocks, making model aggregation particularly challenging due to the variance in features and data across clients. The paper introduces an innovative algorithm for hybrid FL, which assigns a distinct feature extractor to each feature block and integrates their outputs through a classifier. To amalgamate client-specific models, the global model encompasses all feature extractors alongside a unified classifier, which is refined using regularization techniques to minimize discrepancies with individual client models. Experimental results presented by the authors demonstrate the superiority of their algorithm over models that are trained locally.

**Audience:**

Yes

**Broader Impact Concerns:**

No ethical concern

**Claims And Evidence:**

Yes

**Requested Changes:**

Some changes, especially on the experiment part, will not only strengthen the paper's contribution to the field of federated learning but also provide a clearer, more comprehensive understanding of the proposed algorithm's novelty, effectiveness, and applicability in various settings.

1. Comparison with Basic Approaches: The manuscript should include a discussion on how the proposed hybrid federated learning (FL) algorithm compares to more straightforward approaches, such as adapting FedAVG or FedProx for hybrid FL by imputing missing features with zeros. This comparison is crucial for understanding the necessity and advantage of the proposed method over these naive solutions.

2. Detailed Analysis of Model-Matching Algorithm: It is essential to provide a more in-depth analysis of the model-matching algorithm, especially considering its potential computational complexity when dealing with a large number of feature blocks.

3. Expansion of Experimental Settings: To better validate the applicability and robustness of the proposed algorithm, the experimental section needs significant expansion. For instance, incorporating a wider range of $d_0$ and $d_m$ settings in the experiments to comprehensively assess the algorithm's performance across different configurations.

**Strengths And Weaknesses:**

Pros,

1. The exploration of hybrid federated learning (FL) by the authors addresses a novel and significant area that has not been extensively studied. The algorithm proposed is both innovative and demonstrates encouraging outcomes.

2. Additionally, the theoretical convergence of the algorithm has been analyzed, albeit with the reliance on some assumptions that may not be practical.

Cons.

1. A basic approach for hybrid FL could involve adapting existing FL algorithms like FedAVG or FedProx by imputing missing features with zeros. The manuscript would benefit from a discussion on these methods.

2. The paper could be enhanced by a more thorough examination of the model-matching algorithm. Given that this process involves solving optimization problems, its computational complexity could become significant with an increase in the number of feature blocks. An analysis of the implications of simply averaging the models in this context would be valuable.

3. The experimental setup presented in the paper is somewhat restricted. While the problem is introduced with a medical diagnosis scenario involving multi-modal data, the experiments are limited to image classification tasks. A broader range of settings for $d_0$ and $d_m$ would provide a more comprehensive evaluation of the algorithm’s performance, which may vary depending on $d_0$, $d_m$, and the ratio $d_0/d_m$.

4. The explanation of how the proposed algorithm simplifies to FedAvg under certain conditions, specifically when $\mu_1$ is very large, is not sufficiently clear. Clarifying how the client update mechanism in Equation 8 diverges from that of FedAvg under these circumstances would be beneficial.

---

> ### Author Response · Authors · 2024-03-08
> **Reply to reviewer yZ9H**
>
> 1. **Comparing with HFL algorithm:** In appendix C.1, we provide the comparison between HyFEM and the conventional HFL algorithm (FedProx). The results shows that HyFEM can have better performance than HFL algorithm.
> 2. **Detailed analysis of ModelMatching algorithm:** The model matching algorithm uses Hungarian method on the classifier neural matching step, and the algorithm complexity is $O(PM{d}^3_{max})$, where $d_{max}$ is the maximum width of the clients' classifier, $P$ is the number of outer iterations of the model matching algorithm (typically $4-8$), and $M$ is the client number. In practice, the width of the classifier will not be extremely large, typically ranging from $10^3$ to $10^4$ in size. With the parallel implementation of the algorithm (e.g., [R1]), performing one round of the Hungarian algorithm only takes a few seconds, and the total runtime of the model-matching algorithm is around 1-2 minutes. Compared with the runtime of the clients' local forward/backward steps, such computation time is minor. Additionally, in the extreme case where the classifier width or the number of clients is extremely large, one can overlap the ModelMatching algorithm with the ClientUpdate algorithm. That is, at iteration $t$, at the client side, ClientUpdate uses the matching pattern at the previous iteration $\Pi_m^{t-1}$, instead of $\Pi_{m}^t$, and the server side runs ModelMatching to compute $\Pi^{t}_{m}$ for the next iteration in parallel.
> 3. **Expansion of experiment settings:** Thank you for the precious advice. We have conducted additional experiments to study the effect of the regularizers $\mu_2$ in the problem, and we will include the experiment result in the revised manuscript. However, it is hard for us to conduct extra experiments on a wider range of the number of feature blocks $d_0, d_m$ for the following reasons: 1) $d_0$ and $d_m$ are the feature block number, which depends on the data type. For the current datasets (Cifar10, ModelNet40, EuroSAT, HeriGraph), it is hard to further split the data into more feature blocks; 2) In the targeted medical diagnosis scenario, is also impractical to have tens or more feature blocks; 3) scaling to a larger number of feature blocks requires searching for a suitable dataset, completely rewrite to the data loader, model and other code blocks. Given the limited time for response (2 weeks), it is hard to complete such modifications to the experiment.
>
> [R1] Date, K., & Nagi, R. (2016). GPU-accelerated Hungarian algorithms for the linear assignment problem. Parallel Computing, 57, 52-72.

---

### Review · Reviewer_r63e · 2024-02-24

**Summary Of Contributions:**

This paper proposes a framework for hybrid Federated Learning where the data from each party share different sample spaces and feature spaces simultaneously. The server model and each user's local model are divided into feature extractors and classifiers. In each global iteration, each user first performs some perturbed SGD steps to optimize the client model, and then the server aggregates all the client models to update the feature extractors and classifiers separately. The update method is inspired by model matching algorithms.

**Audience:**

Yes

**Broader Impact Concerns:**

N/A.

**Claims And Evidence:**

No

**Requested Changes:**

See the weaknesses.

**Strengths And Weaknesses:**

Strength
1.	This paper provides proof of convergence.
2.	The description of the algorithm is very detailed and easy to follow.
3.	This algorithm makes appropriate improvements to the model matching algorithm.

Weakness
1.	Although hybrid FL is relatively less studied compared to other FL topics, there are still related works. The authors should discuss and compare the proposed algorithm with prior works.
2.	The complexity of the Hungarian algorithm in the model matching algorithm is cubic, which may become a significant overhead. The authors should discuss the efficiency of this method.
3.	The update process for the regularizer does not incorporate weighting for statistical heterogeneity, which could result in poor performance when dealing with statistically heterogeneous data.

---

> ### Author Response · Authors · 2024-03-08
> **Reply to reviewer r63e (Part I)**
>
> 1. **Related Work:**
>     1. Hybrid Federated Learning (Hybrid FL) can refer to different settings in the existing literature. In [R1], it refers to the setting where the training is a hybridization of centralized learning and federated learning, i.e., the training samples are held partially by the server and partially distributed to the clients. In [R2], Hybrid FL for sensor network refers to a hierarchical federated learning system, where each client is consist of multiple sensors with vertically distributed data and the clients are connected to a server to perform horizontal FL for distributed sensing. Although named as hybrid FL, the above-mentioned settings are different from the settings on our paper, i.e., each client holds partial samples and partially overlapping features. Therefore, these algorithms cannot be applied to our setting.
>     2. Federated contrastive learning (FedCL) is another set of algorithms related to Hybrid FL. In this setting, the clients hold non-overlapping features and partially overlapping samples and aim to learn separate models for local inference (e.g., [R3, R4]). The algorithms perform vertical FL on the overlapping samples to train a global guidance model and perform local contrastive learning (self-supervised learning) with the non-overlapped local data to train local models. Compared with our setting, the FedCL algorithm requires overlapping samples and transmitting intermediate features as VFL and fails to make use of overlapping feature spaces and non-overlapping samples during VFL training.
> 3. **Efficiency of Hungarian method:** The model matching algorithm uses Hungarian method on the classifier neural matching step, and the algorithm complexity is $O(PM{d}^3_{max})$, where $d_{max}$ is the maximum width of the clients' classifier, $P$ is the number of outer iterations of the model matching algorithm (typically $4-8$), and $M$ is the client number. In practice, the width of the classifier will not be extremely large, typically ranging from $10^3$ to $10^4$ in size. With the parallel implementation of the algorithm (e.g., [R5]), performing one round of the Hungarian algorithm only takes a few seconds, and the total runtime of the model-matching algorithm is around 1-2 minutes. Compared with the runtime of the clients' local forward/backward steps, such computation time is minor. Moreover, in the extreme case where the classifier width or the number of clients is extremely large, one can overlap the ModelMatching algorithm with the ClientUpdate algorithm. That is, at iteration $t$, at the client side, ClientUpdate uses the matching pattern at the previous iteration $\Pi_m^{t-1}$, instead of $\Pi_{m}^t$, and the server side runs ModelMatching to compute $\Pi^{t}_{m}$ for the next iteration in parallel.

---

> ### Author Response · Authors · 2024-03-08
> **Reply to reviewer r63e (Part II)**
>
> 3. **Dealing with statistical heterogeneity:** Indeed, the ModelMatching algorithm does not explicitly consider the impact of the statistical heterogeneity (i.e., clients have different number/distribution of samples). On one hand, it is non-trivial to derive a clear relationship between the data statistics and the model heterogeneity. Therefore, it is hard to reformulate the model-matching problem to have an explicit dependency on the data heterogeneity. However, the matching procedure can implicitly take data heterogeneity into consideration in the construction of the cost matrix. In eq (20) in the appendix, the cost matrix of the matching problem depends on the weights of the client and server model. Based on the statistics of model parameters (e.g., mean and variance of client and server model), the distance function dist$_1$ and dist$_2$ balance the possibility of matching client neuron to the existing server model or expand the server model, so it can take the data heterogeneity into consideration.
> Additionally, to deal with the statistical heterogeneity, our problem formulation uses a *weighted averaging* over the local loss function in eq (6), where $p_m$ represents the averaging weight of client $m$. When choosing $p_m = \frac{|N_m|}{|N|}$ proportional to the size of the client's dataset, the global problem addresses the statistical heterogeneity issue.
>
> [R1] Elbir, A. M., Coleri, S., Papazafeiropoulos, A. K., Kourtessis, P., & Chatzinotas, S. (2022). A hybrid architecture for federated and centralized learning. IEEE Transactions on Cognitive Communications and Networking, 8(3), 1529-1542.
>
> [R2] Su, L., & Lau, V. K. (2021). Hierarchical federated learning for hybrid data partitioning across multitype sensors. IEEE Internet of Things Journal, 8(13), 10922-10939.
>
> [R3] He, Y., Kang, Y., Zhao, X., Luo, J., Fan, L., Han, Y., & Yang, Q. (2022). A hybrid self-supervised learning framework for vertical federated learning. arXiv preprint arXiv:2208.08934.
>
> [R4] Kang, Y., Liu, Y., & Liang, X. (2022). FedCVT: Semi-supervised vertical federated learning with cross-view training. ACM Transactions on Intelligent Systems and Technology (TIST), 13(4), 1-16.
>
> [R5] Date, K., & Nagi, R. (2016). GPU-accelerated Hungarian algorithms for the linear assignment problem. Parallel Computing, 57, 52-72.

---

### Comment · Action_Editor_GK11 · 2024-02-25
**Discussion phase**

Authors:  In the discussion phase, please consider the comments among the reviewers.

-  First, identify if there are any questions, where you are unsure what is intended or how to respond.  You can ask for clarification among the reviewers.

- Second, respond to reviewers to address their concerns.  In many cases, this will involve proposed modifications to your paper.  This is most effective when you respond with a plan in the comments and in some cases provide a preliminarily updated paper (with color coded changes).  If it will require a bit more time, it is sometimes ok to delay the update to the paper pdf as long as the plan is clear.

Make sure to provide reviewers enough time to provide feedback to your response in case your are misinterpreting the issue, or need to modify your plan.

---

### Decision · Action_Editor_GK11 · 2024-04-24

**Recommendation:** Accept as is

**Comment:**

Please make sure to upload a final version of the paper (blue coloring of text removed).

**Audience:**

The topic of hybrid federated learning is in the scope of TMLR, and the paper presents a new approach that performs well.  Thus it is clearly has some interest for the audience.

**Claims And Evidence:**

Two reviewers agree that the paper has provided a novel perspective and algorithm for hybrid federated learning, and it shows improvement in the empirical evaluation.  The other reviewer had concerns about distinguishing it from a related topic, but I judge that the authors had addressed this in the response.